# *Aspergillus fumigatus* Can Display Persistence to the Fungicidal Drug Voriconazole

Jennifer Scott,[a] Clara Valero,[a,b] Álvaro Mato-López,[c] Ian J. Donaldson,[d] Alejandra Roldán,[c] Harry Chown,[a] Norman Van Rhijn,[a] Rebeca Lobo-Vega,[c] Sara Gago,[a] Takanori Furukawa,[a] Alma Morogovsky,[e] Ronen Ben Ami,[e] Paul Bowyer,[a] Nir Osherov,[e] Thierry Fontaine,[f] Gustavo H. Goldman,[b] Emilia Mellado,[c,g] Michael Bromley,[a] Jorge Amich[a,c]

[a]Manchester Fungal Infection Group, Division of Evolution, Infection, and Genomics, Faculty of Biology, Medicine and Health, University of Manchester, Manchester, United Kingdom
[b]Faculdade de Ciências Farmacêuticas de Ribeirão Preto, Universidade de São Paulo, Ribeirão Preto, Brazil
[c]Mycology Reference Laboratory (Laboratorio de Referencia e Investigación en Micología [LRIM]), National Centre for Microbiology, Instituto de Salud Carlos III (ISCIII), Majadahonda, Madrid, Spain
[d]Bioinformatics Core Facility, Faculty of Biology, Medicine and Health, University of Manchester, Manchester, United Kingdom
[e]Department of Clinical Microbiology and Immunology, Sackler School of Medicine Ramat-Aviv, Tel-Aviv, Sackler Faculty of Medicine, Tel Aviv University, Tel Aviv, Israel
[f]Institut Pasteur, Université de Paris, INRAE, USC2019, Unité Biologie et Pathogénicité Fongiques, Paris, France
[g]CiberInfec ISCIII, CIBER en Enfermedades Infecciosas, Instituto de Salud Carlos III, Majadahonda, Madrid, Spain

Jennifer Scott and Clara Valero contributed equally to this article. They are listed in alphabetical order.

**ABSTRACT** *Aspergillus fumigatus* is a filamentous fungus that can infect the lungs of patients with immunosuppression and/or underlying lung diseases. The mortality associated with chronic and invasive aspergillosis infections remain very high, despite availability of antifungal treatments. In the last decade, there has been a worrisome emergence and spread of resistance to the first-line antifungals, the azoles. The mortality caused by resistant isolates is even higher, and patient management is complicated as the therapeutic options are reduced. Nevertheless, treatment failure is also common in patients infected with azole-susceptible isolates, which can be due to several non-mutually exclusive reasons, such as poor drug absorption. In addition, the phenomena of tolerance or persistence, where susceptible pathogens can survive the action of an antimicrobial for extended periods, have been associated with treatment failure in bacterial infections, and their occurrence in fungal infections already proposed. Here, we demonstrate that some isolates of *A. fumigatus* display persistence to voriconazole. A subpopulation of the persister isolates can survive for extended periods and even grow at low rates in the presence of supra-MIC of voriconazole and seemingly other azoles. Persistence cannot be eradicated with adjuvant drugs or antifungal combinations and seemed to reduce the efficacy of treatment for certain individuals in a *Galleria mellonella* model of infection. Furthermore, persistence implies a distinct transcriptional profile, demonstrating that it is an active response. We propose that azole persistence might be a relevant and underestimated factor that could influence the outcome of infection in human aspergillosis.

**IMPORTANCE** The phenomena of antibacterial tolerance and persistence, where pathogenic microbes can survive for extended periods in the presence of cidal drug concentrations, have received significant attention in the last decade. Several mechanisms of action have been elucidated, and their relevance for treatment failure in bacterial infections demonstrated. In contrast, our knowledge of antifungal tolerance and, in particular, persistence is still very limited. In this study, we have characterized the response of the prominent fungal pathogen *Aspergillus fumigatus* to the first-line therapy antifungal voriconazole. We comprehensively show that some isolates display persistence to this fungicidal antifungal and propose various potential mechanisms of action. In addition, using an alternative model of infection, we provide initial evidence to suggest that persistence

Address correspondence to Jorge Amich, jamich@isciii.es.
The authors declare no conflict of interest. In the past 5 years, Sara Gago has received speaker fees from Gilead Sciences and research grant support from Pfizer. Ronen Ben-Ami has served on advisory boards for Pfizer and Merck and received compensation from Teva and Gilead for lectures. These roles have no relation with the work presented in this article.

may cause treatment failure in some individuals. Therefore, we propose that azole persistence is an important factor to consider and further investigate in *A. fumigatus*.

**KEYWORDS** antifungal persistence, *Aspergillus fumigatus*, drug response, treatment failure, voriconazole

*A*spergillus fumigatus is the most prominent fungal pathogen of the human lung, being the major causative agent of a range of diseases collectively termed aspergillosis (1). The incidence of fatal aspergillosis infections is on the rise due to the increase in the at-risk population (2, 3), including severe COVID-19 patients (4–6). The mortality associated with *A. fumigatus* infections remains unacceptably high, with 38% 5-year mortality rates for chronic pulmonary aspergillosis (7) and ranging from ~35% in early diagnosed and treated patients to nearly 100% if diagnosis is missed or delayed in invasive aspergillosis (8). These number represent the highest mortality of all invasive fungal diseases (9). This situation has become even more worrisome due to the emergence of clinical resistance to existing antifungals, which poses a serious threat to human health (10). Azoles are currently the only U.S. Food and Drug Administration-approved class of mold-active agents that can be administered orally and intravenously, and, accordingly, they are used for both the evidence-based treatment and prevention of aspergillosis diseases (11). However, in the last decade, the number of clinical *A. fumigatus* isolates that are resistant to triazole drugs has increased worldwide, causing serious problems for clinical management, as the therapeutic options are reduced and the mortality rates caused by resistant isolates are higher (7, 12, 13). Furthermore, both in clinical disease and in experimental animal models, it has been observed that treatment failure is common even if the infecting fungal isolate is susceptible to the azole used for treatment (14, 15). There are several possible explanations for treatment failure in these cases, including antifungal tolerance or persistence (16, 17).

In the last decade, there has been a revolution in our understanding of how pathogenic microorganisms withstand the challenge of antimicrobial drugs. Classically, pathogens were described to be either susceptible or resistant to a certain drug. Antimicrobial resistance is due to a genetic feature in the microbe (intrinsic or acquired via mutations or gene transfer) that enables it to grow normally at high concentrations of the agent, above the MIC defined for the antimicrobial based on the clinical breakpoint (18–23). However, in recent years it has become obvious that pathogenic bacteria can withstand the action of drugs by at least three other mechanisms: tolerance, persistence, or heteroresistance. These three phenomena share the feature that are strain specific and can evolve and be potentiated by point mutations and yet differ in other details (16, 24–27). Drug tolerance has been defined as the ability of all cells of a genetically isogenic strain to survive, and even grow at low rates, for extended periods in the presence of drug concentrations that are greater than the MIC (28, 29). In persistence, only a small fraction of the isogenic population (usually <0.01% of the cells) survive, and grow at low rates, for an extended period of time in the presence of supra-MICs of the drug (28, 29). These two phenomena do not imply an increase in the MIC. Finally, heteroresistance is the transient capacity of a microbial subpopulation to increase the MIC in the presence of the drug, due primarily to gene amplification and potentially to adaptation or epigenetic modifications (24, 30–33). All these phenomena have been implicated in treatment failure and relapse in bacterial infections (28, 34).

In fungi, tolerance has been often described as "trailing growth," and its relevance for infection has been mostly disregarded. However, a recent landmark study in *Candida albicans* has characterized tolerance as a distinct, strain-specific feature and provided evidence for its relevance in persistent candidemia (17, 35). In *Aspergillus*, trailing growth is known to be prominent in the presence of caspofungin (36, 37) apparently due to single strain heterogeneity (38). Moreover, at higher concentrations of this drug some isolates can resume normal growth, an effect known as the "Eagle" or paradoxical effect (see review reference 39 for more information), which we have recently demonstrated as a tolerance phenotype (40). The possible relevance of the

paradoxical effect in clinical practice is still under debate, but there is evidence suggesting that it might be important (41), and concern has been raised by some clinicians (42). Remarkably, it seems that the phenomenon of tolerance in fungi is seen with static drugs, e.g., azoles for *C. albicans* and echinocandins for *A. fumigatus*. However, the possibility that *A. fumigatus* can display tolerance or persistence to azole antifungals had not been previously investigated. Interestingly, in contrast to *C. albicans* and *Cryptococcus neoformans*, azoles have been shown to have fungicidal activity against *A. fumigatus* (43–45), which implies that the underlying mechanisms are likely different. While these phenomena have so far only been investigated in bacteria or yeasts, the approaches to detect and investigate azole tolerance or persistence need to be tailored in filamentous fungi, like *A. fumigatus*, as these organisms form multicellular hyphae. Using various complementary approaches, we show here that small subpopulations of certain *A. fumigatus* isolates can survive and even grow at low rates at supra-MICs of the fungicidal drug voriconazole.

## RESULTS

**Some *A. fumigatus* isolates show persistence to voriconazole.** To determine whether *A. fumigatus* can display tolerance or persistence to voriconazole, we examined a collection of isolates consisting of 9 environmental and 10 clinical strains (gifts from Paul Dyer, collection PD-47-XX [here abbreviated as PD-XX]), 5 common laboratory strains, and one resistant control (RC) strain harboring the TR34/L98H mutations in the *cyp51A* locus (see Table S1 at https://doi.org/10.5281/zenodo.7533742), using disc diffusion assays. We evenly spread $4 \times 10^4$ conidia of the isolates on RPMI agar plates, placed a 6-mm disc in the center containing 10 $\mu$L of voriconazole (0.8 mg/mL), and then incubated them for 5 days. We observed variability in the sizes of the inhibition halos, reflecting differences in susceptibility of the isolates (Fig. 1A; see also Fig. S1 in the supplemental material). As expected, the RC strain did not show a proper inhibition halo (Fig. 1A; see also Fig. S1). In contrast, 15 of the 20 isolates showed a clear and well-defined inhibition zone. Interestingly, we found that five isolates were able to form colonies within the halo of inhibition, and four in five of these isolates (PD-9, PD-104, PD-254, and PD-266; see also Fig. S1) formed only a few small colonies, which might be indicative that a few conidia are able to germinate and grow a little in the presence of a supra-MIC of voriconazole (Fig. 1A; see also Fig. S1). The remaining isolate (PD-256) did not show any inhibition halo, suggesting that it is a resistant isolate.

We then assessed the MICs of the original isolates and their derived colonies of the halo (CoHs). A complication of working with a filamentous fungus is that the CoHs need to be grown on a new plate in order to harvest spores (conidia) to be used for MIC determination. We therefore decided to assay two different conidium-harvesting conditions to distinguish the different phenotypes: regrowing the CoHs on solid RPMI in the absence or in the presence of a low concentration of voriconazole (0.12 $\mu$g/mL). It would be expected that for conidia grown in the presence of the drug, both resistance and heteroresistance are detected as an increment in the isolate's MIC. In contrast, for conidia grown in the absence of the drug, the reversible increase in MIC that is characteristic of heteroresistance would be lost, while a stable genetic-based increment in MIC that defines resistance would be maintained. Finally, persistence should not cause a change of the isolate's MIC independently of the presence of the drug in the conidium-harvesting medium. Using conidia obtained from both conditions, the absence or presence of voriconazole, we found that the original isolate which did not show an inhibition halo (PD-256, Fig. 1A; see also Fig. S1) had a very high MIC (>8 $\mu$g/mL), demonstrating that it is a resistant isolate (Table 1). CoHs formed by PD-254 and PD-266, which upon reinoculation did not show an inhibition halo (see Fig. S1), showed an increased MIC compared to the parental isolate (Table 1). The colony picked from PD-254 showed increased MIC when regrown on both media with and without voriconazole, suggesting that the CoH may have acquired a mutation that confers a stable resistance phenotype. In contrast, PD-266, which already had an elevated MIC, only showed an increased MIC when regrown on medium containing voriconazole, suggesting that this strain may be heteroresistant. Finally, CoHs from isolates PD-9 and PD-104

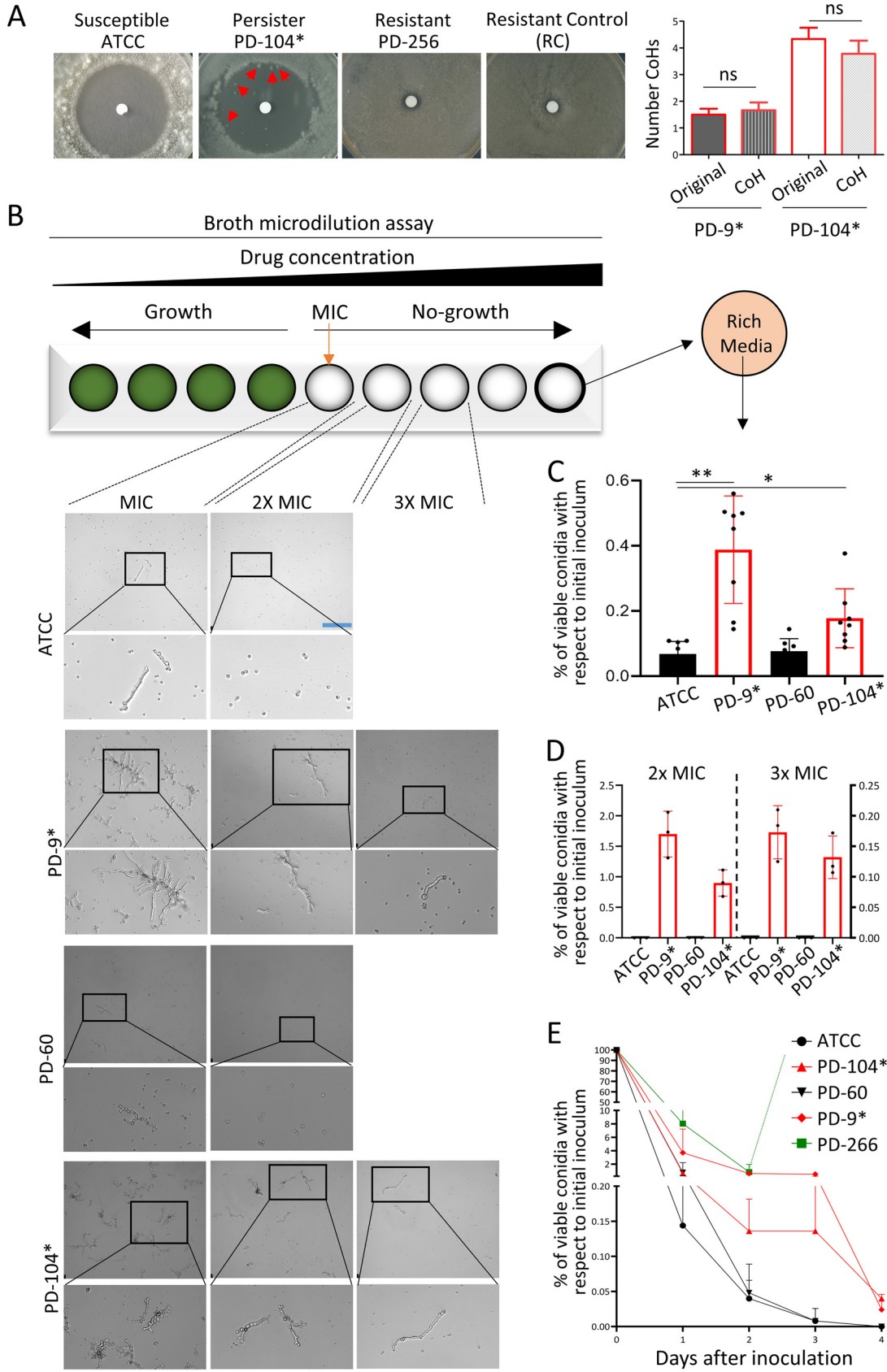

**FIG 1** Certain *A. fumigatus* isolates display persistence to voriconazole. (A) In disc diffusion assays (10 $\mu$L of 0.8 mg/mL voriconazole), the susceptible isolate ATCC never grew any colony in the inhibition halo, and the resistant control (RC) isolate

**TABLE 1** Voriconazole MICs of original isolates and their derived colonies of the halo (CoHs)[a]

| Isolate[a] | Voriconazole MIC ($\mu$g/mL) | | | | CoH classification |
| --- | --- | --- | --- | --- | --- |
| | Original | Original (grown on drug) | CoH (no drug) | CoH (grown on drug) | |
| ATCC | 1 | 2 | | | Susceptible |
| PD-9* | 0.5 | 1 | 0.5 | 1 | Persister |
| PD-104* | 1 | 2 | 1 | 2 | Persister |
| PD-254 | 2 | 2 | >8 | >8 | Resistant |
| PD-256 | >8 | >8 | >8 | >8 | Resistant |
| PD-266 | 4 | 8 | 4 | >8 | Potentially heteroresistant |

[a]Voriconazole MICs of the original isolates and their derived CoH, grown in the absence or presence of low (0.12 $\mu$g/mL) concentrations of voriconazole, as indicated. Susceptibility or resistance was determined based on the MIC calculated from Broth dilution assays, using the breakpoints defined by Guinea (152). *, Persistence.

showed the same MICs as their original isolates after a passage in the absence or presence of voriconazole (Table 1). This suggests that they are not resistant or heteroresistant derivatives of the original isolates. Indeed, repetition of the disc assay with the original isolates and with CoH regrown in the presence of voriconazole, showed a similar level of colony appearance in the original strains and their derived CoHs for PD-9 (expressed as the mean $\pm$ the standard error of the mean: original, 1.5 $\pm$ 0.22; CoH, 1.7 $\pm$ 0.29) and PD-104 (original, 4.3 $\pm$ 0.42; CoH, 3.8 $\pm$ 0.49) (Fig. 1A). In contrast, PD-266 CoH upon reinoculation did not form an inhibition halo, reflecting again a transient increase in its MIC (see Fig. S1). Although no formal measurement of growth rate was performed, the clear difference in size between single persister (PD-9 and PD-104) and potentially heteroresistant colonies (PD-266) after the same time of incubation (see Fig. S1) indicate that the growth rate of persister colonies is reduced. Interestingly, all isolates showed a small increase (one dilution) in MIC when regrown in the presence of drug (Table 1). This may be due to the development of conidia adapted to the growing environment in preparation for the subsequent germination, an effect recently described in *Aspergillus* spp. (46). Next, we tested whether CoHs could also be detected using Etest strips, which is a well-established method to measure the MIC and is thus more quantitative than a disc diffusion assay (47). In agreement with disc diffusion assays, CoHs developed inside the halo of inhibition created by Etests on persister isolates (PD-9 and PD-104), but not on nonpersister isolates (ATCC 46645 = ATCC and PD-60) (see Fig. S2A). Finally, to verify that the persistence phenotype is stable, we sequentially passaged CoHs isolated from PD-9 and PD-104 on potato-dextrose agar (PDA) media (without voriconazole) and performed disc diffusion assays every two or three passages. The halo of the disc and the relative level of persistence were maintained for 10 passages (see Fig. S2B), demonstrating that persistence is a stable phenotype.

In bacteria, persister and tolerant cells are often dormant and do not grow (29). To investigate this in *A. fumigatus* in more detail, we followed a protocol to detect tolerance/persistence in bacteria in which the disc containing the drug, after a period of

**FIG 1** Legend (Continued)

grew up to the edge of the disc, as did the strain PD-256. The persister isolate PD-104 was consistently able to grow a few small colonies. Plates were incubated for 5 days at 37°C. Quantification of the CoHs formed by persister isolates and their derived CoHs is shown. (B) Inspection of the wells of a broth dilution assay under a microscope 72 h after inoculation revealed that the nonpersister isolates ATCC and PD-60 displayed only limited microscopic growth at the MIC, and all conidia remained nongerminated at higher concentrations. In contrast, the persister strains PD-9 and PD-104 showed noticeable microscopic growth up to 3-fold (3×) the MIC. Scale bar, 132.5 $\mu$m. (C) The full content of the well containing the maximum concentration of voriconazole (8 $\mu$g/mL) was plated on rich medium PDA plates, and CFU were counted 48 h after inoculation. Persister isolates grew significantly more CFU than nonpersister isolates (PD-9 versus ATCC 46645C [$P = 0.002$] and PD-104 versus ATCC [$P = 0.0331$]), demonstrating that these strains remain viable upon azole treatment for a longer period. Three independent experiments with three biological replicates were performed. The graph represents the means and standard deviations (SD), and data were analyzed using the Brown-Forsythe and Welch analysis of variance (ANOVA) test with Dunnett's multiple comparisons. (D) Conidia were directly inoculated on RPMI solid plates containing 2× or 3× the MIC of each isolate and incubated for 48 h at 37°C. The graph depicts the percentage of conidia from the original inoculum ($2 \times 10^4$) that were able to form colonies. Three independent experiments were performed. (E) A survival curve in the presence of 4 $\mu$g/mL of voriconazole revealed that, while nonpersister strains lost viability very rapidly, the persister isolates had the characteristic biphasic reduction in viability, showing that a subpopulation of the persister isolates remained viable for a longer period. Two independent experiments with biological duplicates and two technical replicates were performed; the graph represents the means and SD. Persister isolates are labeled with an asterisk (*).

incubation, is switched with another one containing fresh media (48). The rationale is to detect persister cells that survive the action of the drug but do not grow or grow very slowly in its presence. After initial growth we substituted the disc with voriconazole for a new one containing *Aspergillus* minimal medium (AMM). Interestingly, after the disc switch, colonies were observed in the halos for several isolates (CEA10, PD-7, -8, -50, -154, -249, and -264; see Fig. S3), suggesting that these may also be persister isolates. Nevertheless, three isolates—PD-9, -104, and -259—were able to form colonies before the disc switch, and of these three, only PD-9 and PD-104 grew even more colonies after disc switch (see Fig. S3). Therefore, the isolates PD-9 and PD-104 were able to survive and grow at lower rates in the presence of supra-MICs of voriconazole, a phenomenon that complies with the definition of persistence.

To further characterize persistence in *A. fumigatus*, we performed MIC assays with selected strains and checked microscopic growth of persister and nonpersister isolates at supra-MICs. We found that 72 h after inoculation the nonpersister strains ATCC and PD-60 displayed microscopic growth only at the MIC, and no growth at all at higher concentrations (Fig. 1B; see also Fig. S4). In contrast, the persister isolates PD-9 and -104 showed noticeable microscopic growth at 2× and even at 3× MIC germinated conidia could be detected (Fig. 1B; see also Fig. S4A). While the level of growth was variable among independent experiments, the difference between persister and nonpersister isolates was consistently observed (Fig. 1B; see also Fig. S4). In addition, we plated the entire content of the wells containing the highest concentrations of voriconazole (8 $\mu$g/mL) and found that 48 h after inoculation the nonpersister strains were nearly all killed (only ~0.07% of the conidia survived), while in the persister strains a significantly higher ratio of conidia survived (0.39% of PD-9 $P = 0.002$ and 0.18% of PD-104 [$P = 0.0331$]; Fig. 1C). Therefore, a subpopulation of the *A. fumigatus* persister isolates can survive for long periods in the presence of high concentrations of voriconazole, and some conidia can even grow at a low rate, which seem to be inversely correlated with the concentration of drug. Interestingly, this agrees with the previous disc diffusion assays, where CoHs grew only near the edge of the inhibition halo created by voriconazole (=supra-MIC but lower), but when the drug was withdrawn CoHs appeared closer to the disc (=supra-MIC and higher). To clarify this concentration-dependent ability to grow of persister strains, we directly inoculated conidia on RPMI solid plates containing a very high (8 $\mu$g/mL) or lower (2× and 3× MIC) concentrations of voriconazole. Congruently with the previous observations, after 48 h of incubation persister isolates formed many colonies on 2× MIC (1.66% of original inoculum for PD-9 and 0.87% for PD-104) and some on 3× (0.173% of original inoculum for PD-9 and 0.132% for PD-104), while the nonpersister isolates did not (Fig. 1D), and none of the isolates formed visible colonies on the very high concentration of voriconazole (even after 5 days of incubation). Therefore, the persister isolates can survive at very high supra-MICs for extended periods, but active (slow-rate) persister growth can only take place at relatively lower supra-MICs.

In bacteria, a combination of MIC and speed of killing is the best approach to reveal and differentiate tolerance and persistence (28). Therefore, we investigated the dynamics of cell death caused by voriconazole for persister and nonpersister isolates. Conidia from two susceptible (ATCC and PD-60), two persister (PD-9 and PD-104), and potentially heteroresistant (PD-266) strains were incubated for 4 days in liquid RPMI in the presence of 4 $\mu$g/mL of voriconazole and aliquots of the culture plated (after phosphate-buffered saline [PBS] washing to remove the drug) on PDA-rich media every 24 h (Fig. 1E). We found that for all strains the number of viable conidia dramatically declined after 24 h of incubation, reflecting a strong fungicidal action of voriconazole, even against an isolate with high MIC. Interestingly, while the susceptible isolates were completely killed within 48 to 72 h, a considerable number of conidia from the persister strains maintained viability for an extended period (~0.748% for PD-9 and ~0.136% for PD-9 at 48 h and ~0.52% for PD-9 and ~0.128% for PD-9 at 72 h; Fig. 1E), resulting in the characteristic biphasic killing curve that is the hallmark of persistence (29). The isolate PD-266 showed visible growth from 48 h, which agrees with its potential heteroresistant phenotype, and saturated the plate, making it impossible to count the CFU. To investigate the survival of the isolates at the single cell

(conidial) level, we incubated the two persister strains and one susceptible strain in liquid RPMI in the presence of 32 $\mu$g/mL voriconazole for 48 h and replaced it with fresh drug-free medium. We observed that during the first 24 h of incubation after drug withdrawal there was no growth for any of the isolates, but the persister strains were subsequently able to resume growth (see Videos S1 and S2 in the supplemental material at https://doi .org/10.5281/zenodo.7533742), while the susceptible strain was not (see Video S3 at https://doi.org/10.5281/zenodo.7533742). Hence, persister strains indeed can survive for extended periods in the presence of high concentrations of voriconazole and resume growth when the drug is withdrawn.

To check whether the observed phenomena could be explained by mutations in the azole target genes, we sequenced the promoter and open reading frame (ORF) of *cyp51A* and the ORF of *cyp51B* in the persister isolates PD-9 and PD-104 and the nonpersister strain PD-60. In all strains, we found that *cyp51A* was completely wild type (there were no nucleotide changes) and *cyp51B* had only synonymous polymorphisms (see Table S2 at https:// doi.org/10.5281/zenodo.7533742), thus denying an implication of target enzyme mutations in the phenomenon of persistence. We also sequenced the sterol-sensing domain of *hmg1*, as it has been proposed that mutations in this gene may be a precursor step for the development of azole resistance (49, 50), and in bacteria persistence has been shown to correlate with the evolution of resistance (51, 52). However, we found again that all strains harbored a completely wild-type sequence (not shown), suggesting that *hmg1* is not related with voriconazole persistence.

In conclusion, we have observed that certain *A. fumigatus* isolates can survive and grow at low rates for extended periods of time in the presence of supra-MICs of voriconazole; therefore, these isolates display persistence to voriconazole.

***A. fumigatus* persistence is maintained in the hyphal transition and seems to be dependent on the nutritional environment.** *A. fumigatus* undergoes morphological changes during its development, shifting from resting conidia to swollen conidia (~4–5 h), then to germlings (~8 h), and finally forming hyphae (~16 h) (see Fig. S5A). Since the cellular metabolic status is different at each developmental stage (53–55), we speculated that it might influence the capacity of the fungus to deploy persistence in the presence of voriconazole. Initially, to rule out that persistence could be explained by basal differences in germination or growth, we examined the germination and growth rates of two persister (PD-9 and PD-104) and two nonpersister (ATCC and PD-60) isolates. We found that all isolates germinated at equivalent rate and ratio (see Fig. S5B) and grew similarly on both solid (see Fig. S5C) and liquid (see Fig. S5D) RPMI media. Once excluded differences in germination or growth, we tested the influence of the morphological stage on persistence by incubating the RPMI plates inoculated with the fungus for 8 or 16 h before adding the drug to the disc. We found that the persister strains PD-9 and PD-104 were also able to form colonies in the halo when drug was added 8 h after the beginning of incubation (Fig. 2A), suggesting that persistence is not determined by the developmental stage. We could not draw definitive conclusions for the 16-h-grown hyphae, since the background growth was too dense to undoubtedly differentiate specific persister colonies (Fig. 2A). Nevertheless, we could clearly detect a growing colony in the voriconazole halo for the strain PD-104 (Fig. 2A), proving that at least this strain can display persister growth when short hyphae are challenged with the drug. Interestingly, it has previously been reported that morphological status alters *A. fumigatus* susceptibility profiles against various drugs, including voriconazole (56), which supports the notion that persistence is a different phenomenon, triggered by distinct mechanisms, and is independent of the MIC.

The nutritional environment has been shown to influence tolerance in both *C. albicans* (35) and in *S. cerevisiae* (57). To determine whether *A. fumigatus* persistence is also influenced by the availability of nutrients, we performed a voriconazole disc diffusion assay with the PD-104 isolate and the wild-type control ATCC on rich medium (PDA) or minimal medium (AMM). The PD-104 strain showed persister growth on both media, forming similar numbers of CoHs on AMM (4.0 $\pm$ 0.40) and slightly increased on PDA-rich media (5.5 $\pm$ 0.87), although this was not significant ($P = 0.214$, unpaired $t$ test)

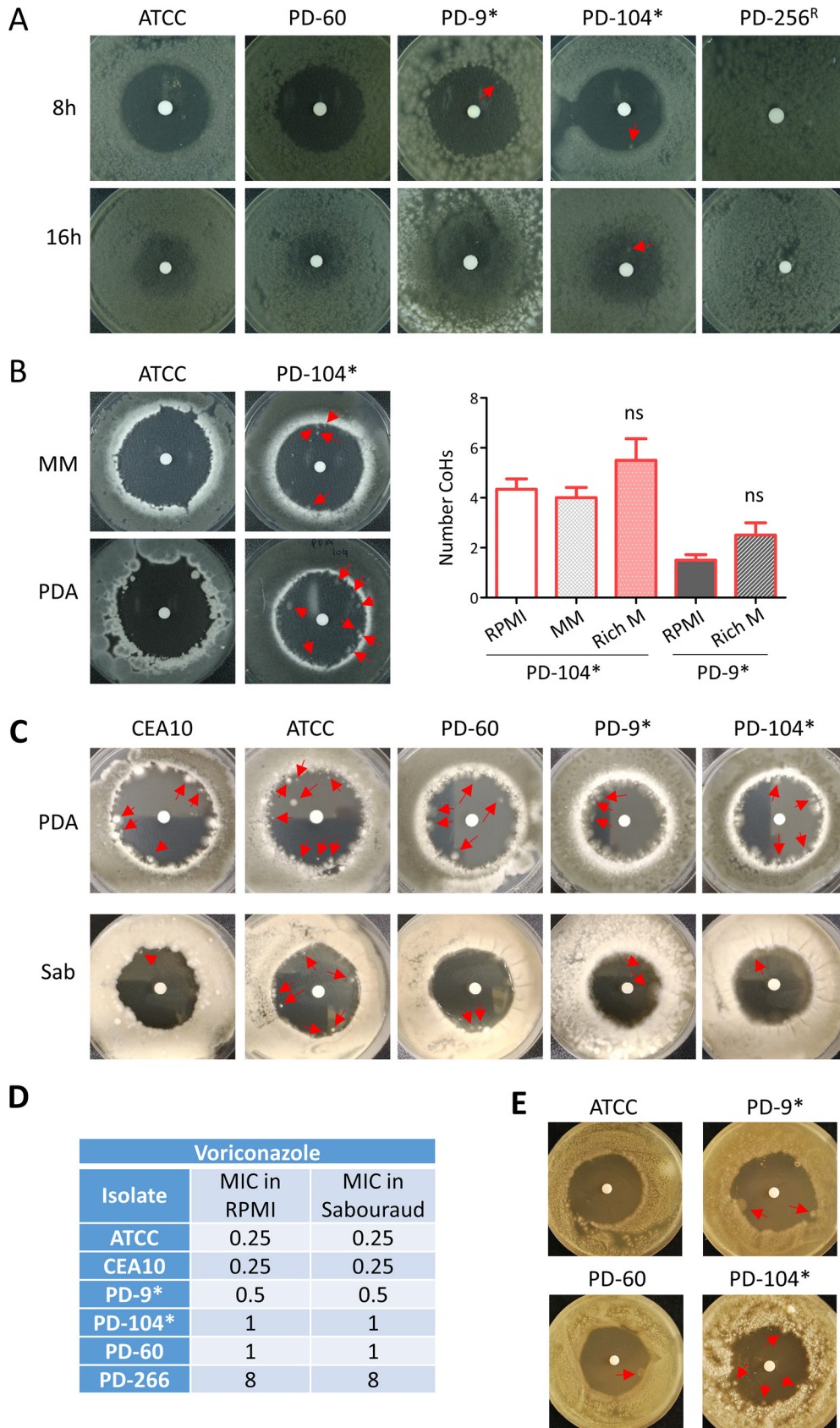

**FIG 2** *A. fumigatus* persistence does not depend on the morphological stage but seems to be determined by the growth media. (A) Conidia of the different isolates were inoculated and incubated for 8 or 16 h before the disc

(Fig. 2B). The ATCC wild-type strain did not show persistence on AMM, but, surprisingly, it did form colonies in the halo on PDA media (Fig. 2B). To investigate this in more detail, we repeated the disc assay with more strains using two different rich media: PDA and Sabouraud (Fig. 2C). The nonpersister isolates CEA10 and PD-60 were also able to form colonies of the halo, and the persister PD-9 seemed to slightly increase the number of CoHs as well although this increment was again not significant ($2.5 \pm 0.5$, $P = 0.078$ [unpaired $t$ test]; Fig. 2B). The MICs of the isolates, measured by broth dilution assay, were equal on Sabouraud and RPMI media (Fig. 2D), indicating that the effect of the drug is not reduced in rich medium. Therefore, the ability of *A. fumigatus* to display persistence seems to be medium dependent, suggesting that a rich nutrient environment favors survival at supra-MICs. Nevertheless, it should also be considered that voriconazole diffusion may be affected in solid rich media, which could be confounding this observation.

Finally, we also considered that persistence might be affected by the age of the spores (all experiments are performed with freshly harvested spores). To check this, we performed disc diffusion assays using 10-week-old spores. We did not observe remarkable differences with respect to fresh spores, although in most of the experiments we could see one colony in the halo of the nonpersister PD-60 (Fig. 2E), suggesting that aged spores might have a slightly higher persistence potential, at least for some isolates.

***A. fumigatus* persistence to voriconazole is independent of stress and cannot be inhibited with adjuvant or antifungal drugs.** Changing environmental conditions activate signaling cascades that trigger transcriptional adaptation and cell wall alterations (58–61). Therefore, we reasoned that the capacity of certain isolates to survive and grow in supra-MICs of voriconazole might be influenced by environmental stressors. Indeed, in *C. albicans* mutants and inhibitors of stress response pathways eliminate tolerance (62). To investigate this possibility, we analyzed voriconazole persistence of PD-104 in the presence of hypoxic (1% $O_2$), oxidative (0.01% $H_2O_2$), osmotic (150 mM NaCl), membrane (0.05% sodium dodecyl sulfate [SDS]), and cell wall (10 $\mu$g/mL CalcoFluor White) stresses. Surprisingly, in contrast to *C. albicans* (35), we found that most environmental conditions did not influence persistence in *A. fumigatus* (Fig. 3A). This suggests that the underlying mechanism(s) of persistence in *A. fumigatus* is likely different from the previously proposed mechanisms of tolerance in *C. albicans*. The only conditions that influenced *A. fumigatus* persistence were SDS, which significantly reduced the number of CoHs ($1.25 \pm 0.47$, $P = 0.0014$ [unpaired $t$ test]; Fig. 3B), and hypoxia, which eradicated persistence (Fig. 3A and B). Since we had observed above that persistence was influenced by the growth medium, we wondered whether hypoxia could also prevent persistence on rich medium. As shown in Fig. 2B and C, all isolates were able to grow colonies in the halo when grown on the rich yeast extract glucose (YAG) medium under normoxia (Fig. 3B). However, persistence was eliminated under hypoxia for all strains, except for PD-104, for which it was strongly and significantly reduced ($0.33 \pm 0.19$, $P = 0.0005$ [unpaired $t$ test]; Fig. 3B) but not completely eradicated. Therefore, it seems that hypoxia reduces persistence, but the nutritional composition of the medium also influences its impact on the phenomenon.

In *C. albicans*, fluconazole tolerance (but not resistance) can be prevented with the use of adjuvant drugs that block general stress signaling pathways (35). To further investigate whether the underlying mechanism of persistence may be different in *A. fumigatus*, we

**FIG 2** Legend (Continued)
containing voriconazole (10 $\mu$L of 0.8 mg/mL) was added to the RPMI plate. At 8 h, when conidia have germinated, the persister isolates PD-9 and PD-104 were still able to grow small colonies in the inhibition halo, while the nonpersister strains were not. At 16 h, when conidia have already formed hyphae, the background growth made impossible to distinguish colonies in PD-9, but a clear one could be detected in PD-104. (B) On *Aspergillus* minimal medium (MM), the persister isolate PD-104 was able to form colonies in the halo, but the nonpersister strain ATCC was not. Quantification of colonies formed my persister isolates on rich media is shown. (C) On rich PDA and Sabouraud (Sab) media all isolates were able to form conidia in the halo. (D) This happened even the MIC of all isolates was the same in commonly assayed using RPMI and rich Sabouraud media. (E) Ten-week-old spores displayed a similar persister phenotype as freshly isolated spores. Often, a single colony could be detected in the halo of inhibition on the nonpersister PD-60, suggesting that aged spores might have slightly more persistence capacity. Persister isolates are labeled with an asterisk (*).

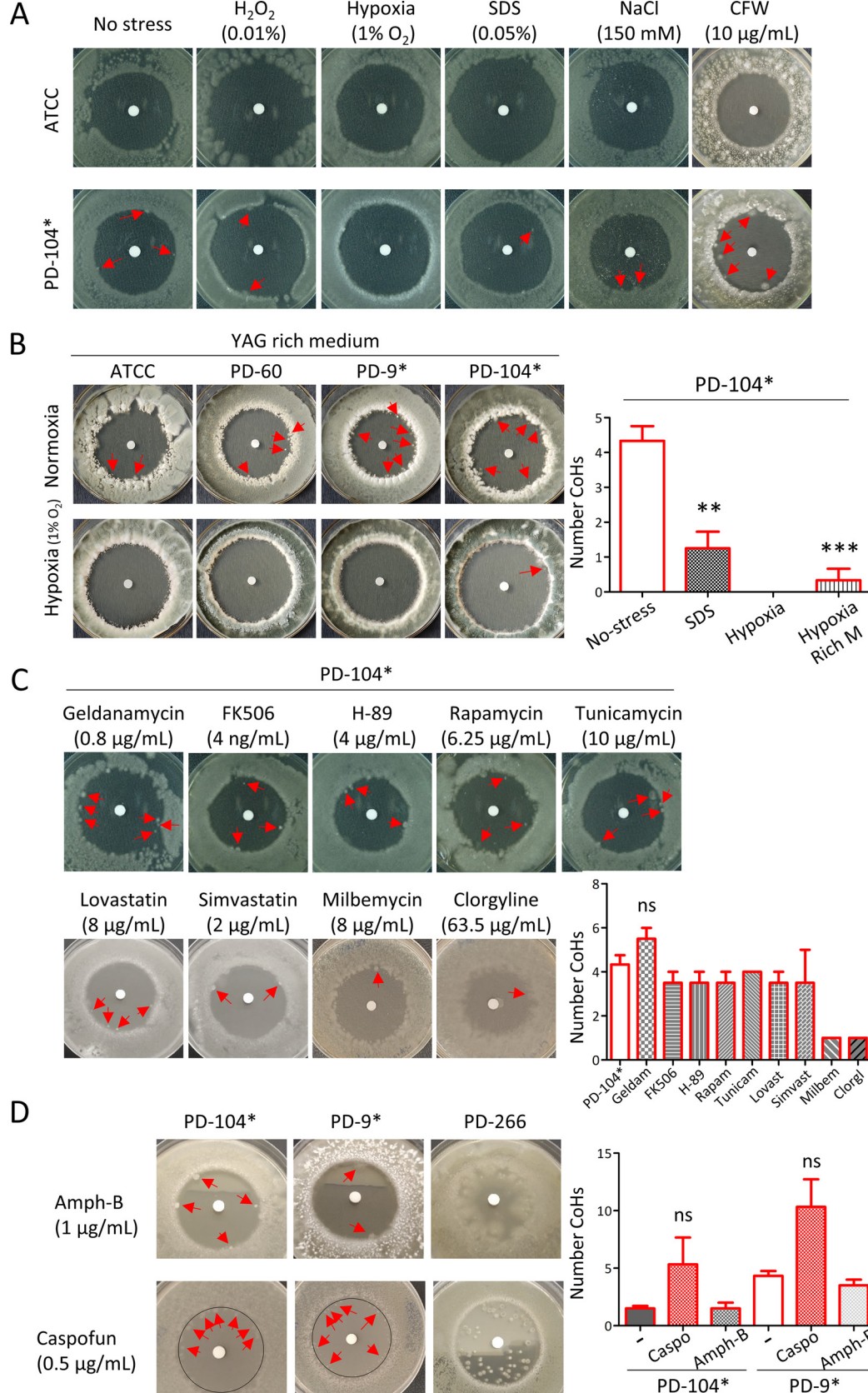

**FIG 3** *A. fumigatus* persistence is not influenced by stress, with the exception of hypoxia, and cannot be eliminated with adjuvant or combinatorial treatments. (A) Hypoxia was the only stress that could prevent persistence, while neither

tested the effect of various drug adjuvants that were previously shown to eliminate tolerance in *C. albicans* (35): geldanamycin (0.8 $\mu$g/mL), an inhibitor of heat shock protein (Hsp90) (63, 64); FK506 (4 ng/mL), an inhibitor of calcineurin (65); H-89 (4 $\mu$g/mL), an inhibitor of the cAMP-dependent protein kinase A (66); rapamycin (6.25 $\mu$g/mL), an inhibitor of the mammalian target of rapamycin (mTOR) (67); and tunicamycin (10 $\mu$g/mL), an inducer of the unfolded protein response pathway (68). In contrast to *C. albicans*, the use of adjuvant drugs did not prevent persistence of *A. fumigatus* isolates (Fig. 3C). Most of these drugs did not seem to influence persistence, except geldanamycin, which seemed to slightly increase it, although it was not significant (5.5 $\pm$ 0.50, $P = 0.197$ [unpaired *t* test]; Fig. 3C). We also tested lovastatin (8 $\mu$g/mL) and simvastatin (2 $\mu$g/mL), since statins inhibit 3-hydroxy-3-methylglutaryl-coenzyme A (HmgA) (69), an enzyme in the same metabolic pathway as the target of azoles, and these two were previously shown to have antifungal activity against *Aspergillus* spp. (70). However, statins were also not able to prevent or noticeably influence persister growth (Fig. 3C). Finally, since efflux of antifungals has been proposed to play a role in *C. albicans* tolerance (62), we tested whether the efflux inhibitors milbemycin A oxim (8 $\mu$g/mL) (71) or clorgyline (63.5 $\mu$g/mL) (72) can prevent *A. fumigatus* persistence. These compounds seemed to be able to diminish the persister capacity of the strains, since only one CoH per plate could be detected, and these colonies were exactly at the edge of the halo (Fig. 3C) (the significance could not be tested because all plates had a value of 1, which prohibits statistical analysis). In conclusion, adjuvant drugs cannot prevent *A. fumigatus* persistence to voriconazole, and efflux inhibitors may affect this process and deserve further study.

Next, we evaluated whether persistence can be eradicated using combinatorial treatment with the other classes of antifungal drugs in clinical use. However, neither amphotericin-B (1 $\mu$g/mL) nor caspofungin (0.5 $\mu$g/mL) could prevent persistence (Fig. 3D). Indeed, caspofungin seemed to increase persistence, although the differences were not significant due to the high variability (PD-9, 5.3 $\pm$ 2.3, $P = 0.24$; and PD-104, 10.3 $\pm$ 2.4, $P = 0.13$ [unpaired *t* test with Welch's correction]). Interestingly, these antifungal drugs could also not impede growth in the halo of the potentially heteroresistant strain PD-266 (Fig. 3D). Therefore, it seems that combinatorial treatment with other antifungals cannot prevent persistence in *A. fumigatus*.

**The phenomenon of persistence can be observed with other azole drugs.** To check whether *A. fumigatus* can display persistence in the presence of other azoles, we first employed the disc diffusion assay adding 10 $\mu$L of a 3.2-mg/mL itraconazole solution. We found that the isolates PD-104 and PD-266 were able to form colonies in the halo (see Fig. S6). Upon reinoculation of a CoH (regrown in itraconazole-containing media), the isolate PD-104 showed an inhibition halo of the same size and displayed similar level of CoH appearance, whereas the PD-266 isolate was able to grow on the whole plate and did not show any inhibition halo. Therefore, this suggests that, as observed with voriconazole, isolate PD-104 is persistent and isolate PD-266 is potentially heteroresistant to itraconazole. We then performed a broth dilution assay with our four well-characterized isolates (nonpersisters ATCC and PD-60 and persisters PD-9 and PD-104) to calculate the MIC for itraconazole and isavuconazole and looked under the microscope at supra-MICs (see Fig. S7A and B). We found that the isolates PD-9 and PD-104 showed slight growth at 2$\times$ MIC, while ATCC and PD-60 did not (see Fig. S7B). Moreover, we inoculated the entire content of wells containing the highest concentration of the drugs (8 $\mu$g/mL) on PDA plates and found that a detectable number

**FIG 3** Legend (Continued)
oxidative ($H_2O_2$), cell wall (SDS or CFW), nor osmotic (NaCl) stress significantly influenced persistence. (B) Hypoxia completely eradicated persistence also on the rich medium YAG for ATCC, PD-60, and PD-9, but only reduced persistence for PD-104. (C) The use of adjuvant drugs could not prevent persistence. Quantification of the CoHs formed by PD-104 under each treatment is shown. (D) Combinatorial treatment with amphotericin B and caspofungin did not prevent persistence. Quantification of the CoH formed by PD-9 and PD-104 under combinatorial treatment is shown. All plates were incubated with 10 $\mu$L of 0.8 mg/mL voriconazole added to the disc and the specific condition (stress, adjuvant, or combinatorial drug), as described in the text, present in the medium. Plates were incubated for 5 days at 37°C. All plates and conditions were repeated in at least two independent experiments. Persister isolates are labeled with an asterisk (*).

of conidia from the isolates PD-9 and PD-104 remained viable after 48 h of incubation in the presence of the azoles (see Fig. S7C). Hence, even if more experiments need to be performed for a detailed characterization of persistence to these azoles, these results suggest that the same isolates that are persisters to voriconazole could also display persistence to itraconazole and isavuconazole.

**The persister growth transcriptome suggests that galactosaminogalactan and high expression of sterol biosynthetic genes and exporters may be relevant to establish persistence.** To better understand the heterogeneous nature of persistence, we compared the transcriptome of PD-104 grown under conditions favoring persistence (above the MIC), sub-MIC and in the absence of drug. We inoculated the spores on top of a nylon membrane placed on an RPMI solid plate. We put the disc with voriconazole (0.8 mg/mL) on the membrane and incubated the sample for 5 days at 37°C. Standard halos of inhibition formed on the membrane, and persister colonies appeared for the PD-104 strain but not for the nonpersister A1160 strain (see Table S1), as expected (see Fig. S8A). We harvested mycelium (avoiding conidia as much as possible) from plates without a voriconazole disc ("No Drug"), from the area equidistant from the border of the plate and the inhibition halo ("Low Drug"), and from the colonies in the halo ("Persister") (Fig. 4A). We had to combine persister colonies from 20 plates per replicate in order to obtain sufficient material for RNA extraction. In addition, we harvested mycelia from A1160 (which genome is well annotated, in contrast to ATCC 46645) using No Drug and Low Drug under the same conditions. We performed transcriptome sequencing (RNA-seq) of two biological replicates/condition and compared the transcriptomes of the following conditions: (i) A1160 Low Drug versus No Drug; (ii) PD-104 Low Drug versus No Drug; (iii) PD-104 Persistence versus No Drug; and (iv) PD-104 Persistence versus Low Drug. We considered as differentially expressed genes (DEGs) those with a $\log_2$ fold change of >1 or <−1 and a false discovery rate (FDR) of <0.05.

In A1160, we detected 62 genes significantly upregulated in the presence of low voriconazole concentrations and only 4 that were downregulated (Fig. 4A; see also Data Set S1 at https://doi.org/10.5281/zenodo.7533742). Such a low number of DEGs may be due to the relatively low concentration of voriconazole in the harvested area. Nevertheless, among the upregulated genes we found 10 related to sterol biosynthesis (see Data Set S1 in the supplemental material), including *cyp51A* and *cyp51B*, the expression of which have been reported to increase upon azole challenge (73, 74). Surprisingly, only 21 of the 62 upregulated genes overlapped with our previously published data set, in which 1,492 genes were upregulated upon challenge of A1160 with itraconazole (75). This striking difference may be due to the different drug employed, the different concentrations assayed, and/or the use of solid medium. Interestingly, the 41 DEGs that are specific to this analysis (which can be found in Data Set S1 at https://doi.org/10.5281/zenodo.7533742) contain *cyp51A*, *cyp51B*, and various other genes of the sterol pathway.

For PD-104, the Low Drug versus No Drug comparison retrieved 32 genes upregulated and only 1 downregulated (Fig. 4; see also Data Set S2 at https://doi.org/10.5281/zenodo.7533742). Within the upregulated genes we again found eight genes related with sterol biosynthesis, including *cyp51A*. In addition, nine genes did not overlap with the comparison of A1160 (Fig. 4A; see also Data Set S2 at https://doi.org/10.5281/zenodo.7533742). These differences with A1160 may point to potential strain differences in response to voriconazole. In the Persister versus No Drug comparison, there were 802 upregulated and 249 downregulated genes, and in the Persister versus Low Drug comparison, there were 646 upregulated and 187 downregulated genes (Fig. 4A; see also Data Set S2). When comparing these three analyses, only 53 genes were exclusively upregulated and 39 downregulated in Persister versus Low Drug comparison (detected using BioVenn [76]), suggesting that they may be important to establish the persister phenotype and not only as a response to the drug (Fig. 4A; see also Data Set S2). We performed Gene Ontology (GO) enrichment analyses (on FungiDataBase [77]) for these 53 DEGs, which retrieved a number of biological processes and molecular functions for up- and downregulated genes. It is important to note that the FDR did not reach significance for any of them, which is possibly due to the low number of

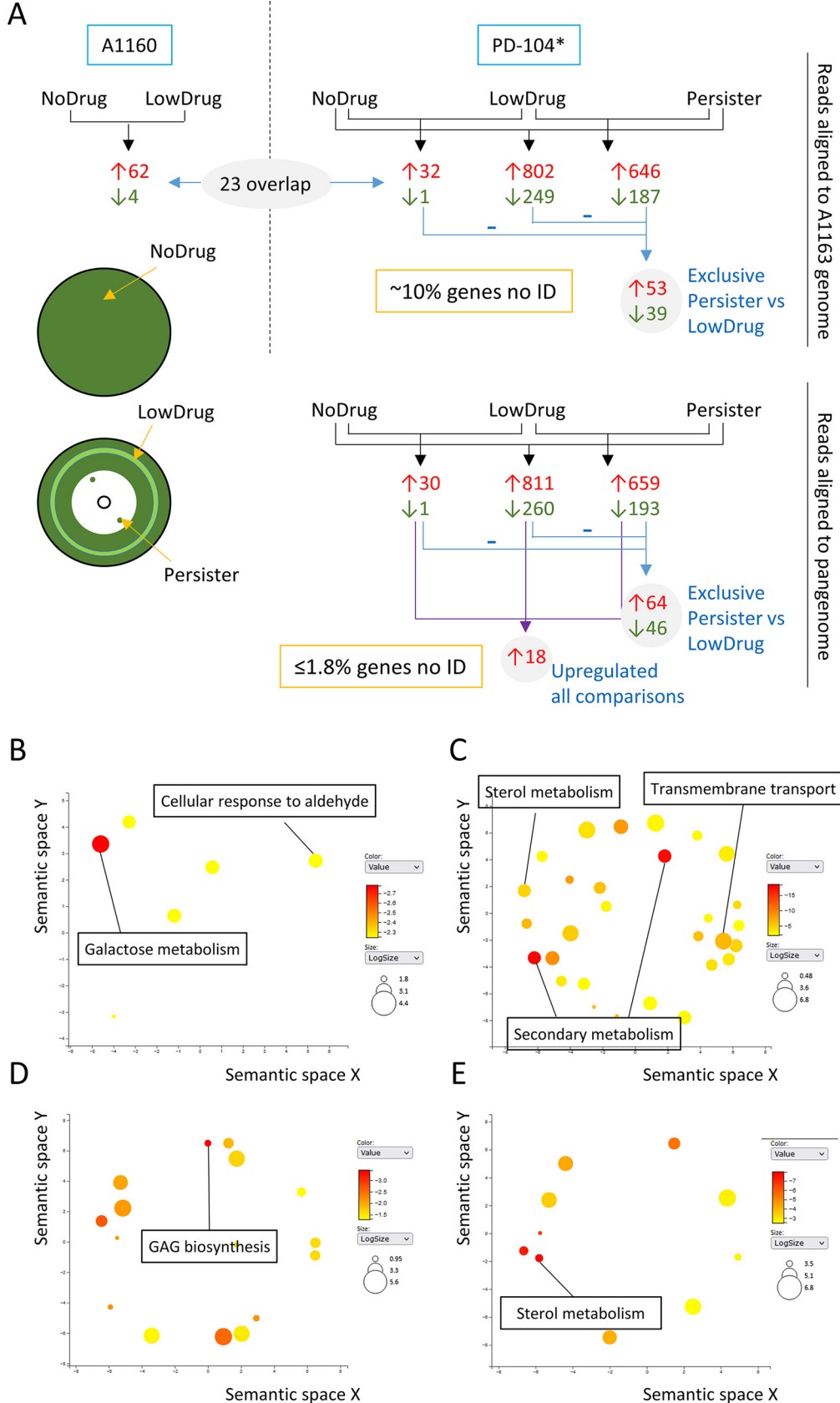

**FIG 4** RNA-seq analysis reveals potential causes of persistence. (A) Schematic representation of the conditions assayed, the comparisons made, and the number of genes identified, as detailed in the text. In addition, the sites of

genes included in the analysis. However, we believe this analysis provided interesting clues and can help direct future research. For the upregulated genes, the strongest enrichment was for biological processes, i.e., galactose and aldehyde metabolism (Fig. 4B), and the molecular function oxidoreductase activity (see Table S3 at https://doi.org/10.5281/zenodo.7533742). Interestingly, galactose metabolism was found to be the most upregulated function in *Staphylococcus aureus* persisters, although the reason has not been elucidated yet (78). For the downregulated genes, there was an enrichment of biological processes related with development and cell cycle (see Table S3 at https://doi.org/10.5281/zenodo.7533742), possibly reflecting the reduced growth rate under this condition.

Interestingly, a significant number of the detected differentially expressed mRNAs in PD-104 were not assigned to any gene of the A1163 genome reference used for the mapping of the reads. In detail, 80 of the upregulated (including 4 of the 20 most upregulated) and 29 of the downregulated for the Persister versus No Drug comparison and 55 upregulated (including 5 of the 10 most upregulated) and 14 downregulated for the Persister versus no Low Drug comparison seem to have no orthologue in A1163 (see Data Set S2). This points to the requirement for a high number (~10% of the DEGs) of strain-specific genes to facilitate persister growth. Therefore, we realigned the sequenced mRNAs from the PD-104 isolate to the newly generated *A. fumigatus* pangenome (79) and performed the comparisons again (see Data Set S3 at https://doi.org/10.5281/zenodo.7533742). In this analysis, the Persister versus No Drug comparison retrieved 811 up- and 260 downregulated genes, and the Persister versus Low Drug comparison retrieved 659 up- and 193 downregulated genes (Fig. 4A; see also Data Set S3). The RNA-seq data were validated by reverse transcription-PCR (RT-PCR) of six selected DEGs (see Fig. S8B). The ratio of identified genes greatly improved, since only ≤1.8% of the DEGs could not be assigned to an ID. Yet, the most upregulated gene in the Persister versus Low Drug comparison was an unidentified gene, and many of the newly identified genes encode uncharacterized hypothetical proteins, which reflects our ignorance of *A. fumigatus* biology and thus how difficult it is to unravel underlying mechanisms in this pathogen. GO enrichment analysis of all DEGs showed that secondary metabolism, ergosterol metabolism, and transport were key upregulated biological processes (Fig. 4C) and that oxidoreductase, catalysis, and transmembrane transport were important molecular functions (see Table S4 at https://doi.org/10.5281/zenodo.7533742), all of which suggests an active metabolic response under persister conditions. As observed above (see Table S3 at https://doi.org/10.5281/zenodo.7533742), downregulated biological processes reflected a downregulation of developmental processes (see Table S4 at https://doi.org/10.5281/zenodo.7533742). Comparison of the DEGs present in all three comparisons revealed that 64 genes were exclusively upregulated and 46 were downregulated in Persister versus Low Drug comparisons (detected using BioVenn [76]), suggesting that they may be specifically important for persistence (Fig. 4A; see also Data Set S3 at https://doi.org/10.5281/zenodo.7533742). GO enrichment analysis of those genes revealed galactosaminogalactan (GAG) biosynthesis as the most significant upregulated biological process (Fig. 4D; see also Table S5 at https://doi.org/10.5281/zenodo.7533742), providing a plausible explanation for the upregulation of galactose metabolism observed before and suggesting that this exopolysaccharide may be relevant for persistence in *A. fumigatus*. Next, we performed a protein functional association analysis using the STRING database and platform (80) to investigate whether these 64 upregulated genes are functionally correlated. A total of 61 of these genes could be matched to proteins in the database, showing a significant ($P = 1.84\mathrm{e}{-}08$) interaction network, includ-

**FIG 4** Legend (Continued)

sampling for the No Drug (plate without voriconazole disc), Low Drug (light green circle at middle of the inhibition halo), and Persister (colonies inside the inhibition halos) conditions are shown. (B to E) The most significantly upregulated GO biological processes are shown using the REVINGO tool for GO data visualization (151). The full list of upregulated GO terms can be found in Tables S2 to S5. Results are shown for PD-104 persister-only genes (aligned to the A1163 genome) (B), all genes upregulated in persistence (aligned to the pangenome) (C), PD-104 persister-only genes (aligned to the pangenome) (D), and genes that are most upregulated in persistence (E) compared to normal response to the drug. Persister isolates are labeled with an asterisk (*).

**TABLE 2** List of 18 genes that showed higher expression in Persistence compared to both No Drug and Low Drug

| Gene ID | Product description |
| --- | --- |
| AFUA_1G03200 | Putative major facilitator superfamily transporter |
| AFUA_4G06890 | 14-$\alpha$ sterol demethylase Cyp51A |
| AFUA_2G00320 | Putative sterol $\delta$-5,6-desaturase |
| AFUA_3G00810 | Putative cholesterol $\delta$-isomerase |
| AFUA_1G03150 | C-14 sterol reductase, putative |
| AFUA_8G02440 | C-4 methyl sterol oxidase, putative |
| AFUA_6G14140 | Has domain(s) with predicted role in response to stress and integral component of membrane localization |
| AFUA_3G00150 | Has domain(s) with predicted oxidoreductase activity and role in metabolic process |
| AFUA_4G01440 | Predicted glutathione *S*-transferase |
| AFUA_4G04820 | C-4 methyl sterol oxidase Erg25, putative |
| AFUA_5G00840 | Ortholog of *A. nidulans* FGSC A4: AN5639, AN2587, AN9444, and AN7395 |
| AFUA_2G15130 | ABC multidrug transporter A-2, putative |
| AFUA_5G03290 | Ortholog of *A. nidulans* FGSC A4: AN819 |
| AFUA_3G02520 | Has domain(s) with predicted role in transmembrane transport and integral component of membrane localization |
| AFUA_7G04740 | Ortholog of *A. fumigatus* Af293: Afu1g01960, Afu3g01040, Afu3g03315, Afu7g01960, Afu8g05750, Afu5g00135, and Afu7g06526 |
| AFUA_2G01890 | Ortholog(s) have 2-octoprenyl-3-methyl-6-methoxy-1,4-benzoquinone hydroxylase activity |
| AFUA_1G14330 | Azole transporter |
| AFUA_8G00710 | Has domain(s) with predicted role in defense response, negative regulation of growth |

ing a node of 17 proteins involved in metabolism (see Fig. S9A and Table S6 at https://doi.org/10.5281/zenodo.7533742). GO analysis of these 17 proteins revealed enrichments in small molecule catabolic process ($P$ = 5.12e–4, FDR = 3.17e–2), D-galactonate catabolism ($P$ = 1.73e–3, FDR = 3.17e–2), and cellular detoxification of aldehyde ($P$ = 3.46e–4, FDR = 3.25e–2), which, given that the proteins in the same node are biologically related, suggests an active detoxification metabolic response during persister growth.

Bacterial persistence has been proposed to be a subpopulation event due to stochastic high expression of relevant genes (81, 82). Accordingly, we reasoned that the genes that are upregulated in Low Drug versus No Drug and Persister versus No Drug comparisons, but also in Persister versus Low Drug comparisons, might reveal genes that are important to adapt to presence of the drug but also that can create persistence with higher levels of expression. We identified 18 genes that appeared as upregulated in all three comparisons (Fig. 4A and Table 2), indicating that they have higher levels of expression in persistence versus normal response to the drug. GO enrichment analysis revealed sterol metabolism as the most significantly upregulated biological process (Fig. 4E; see also Table S7 at https://doi.org/10.5281/zenodo.7533742). Indeed, these few genes had a very significant enrichment in the KEGG pathway "steroid biosynthesis" (Table 2), suggesting that high expression of genes in the sterol biosynthetic route (including *cyp51A*) may enable the subpopulation of persisters to survive and grow in supra-MICs. In addition, these highly expressed genes were enriched in KEGG pathways related with cytochrome P450-dependent drug metabolism (Table 3), which may indicate that detoxification of azoles is also important for persistence. Finally, expression of the *cdr1B* transporter (AFUA_1G14330), known to be associated with azole resistance (83), was also detected to be higher in persistence (Table 2). This suggests that a higher capacity to efflux azoles may also be important for the persister phenotype. Finally, we performed a protein functional association analysis with these 18 genes using the STRING database and platform (80) to search for functional correlations. All 18 genes were matched to proteins and a significant interaction ($P < 1.0e–16$) was found, involving a highly interactive nodule of six proteins related with steroid biosynthesis (see Fig. S9B and Table S8 at https://doi.org/10.5281/zenodo.7533742), suggesting again that a high production of ergosterol is important for persistence.

**TABLE 3** KEGG pathways enrichment of genes upregulated in Persistence compared to Low Drug and No Drug

| KEGG pathway | Name | $P$ | Benjamini |
| --- | --- | --- | --- |
| ec00100__PK__KEGG | Steroid biosynthesis | 2,31E–08 | 1,18E–06 |
| ec00980__PK__KEGG | Metabolism of xenobiotics by cytochrome P450 | 0.002873 | 0.07326 |
| c00982__PK__KEGG | Drug metabolism: cytochrome P450 | 0.004423 | 0.07520 |

**Galactosaminogalactan potentiates persistence of PD-104.** Our transcriptome analysis detected a significantly higher expression of two GAG biosynthetic genes (*sph3*, AFUA_3G07900; and *uge3*, AFUA_3G07910) in persister growth (see Data Set S3 at https://doi.org/10.5281/zenodo.7533742). Manual inspection of the GAG biosynthetic genes (84, 85) revealed that another gene (*agd3*, AFUA_3G07870) might also be upregulated (fold change = 1.01), but although the $P$ value was significant ($P$ = 0.0179), the FDR rate was above the threshold (FDR = 0.1167) (see Data Set S3 at https://doi.org/10.5281/zenodo.7533742). Indeed, we corroborated the higher expression of these three genes in persistence by RT-PCR (see Fig. S8B). This prompted us to investigate whether a larger amount of GAG could create and/or potentiate persistence in *A. fumigatus*. To this aim, we performed broth dilution assays in which we inoculated each isolate (ATCC, PD-9, PD-60, and PD-104) in two different lines and added purified GAG at a concentration of 100 $\mu$g/mL to one of them. Comparison of the MIC with or without GAG demonstrated that this polysaccharide did not affect the resistance profile of the isolates (Fig. 5A). After reading the MIC at 48 h, we added 10 $\mu$g/mL of the dye calcofluor white (CFW) to the wells and imaged the entire well (87 photos using a 20× objective) at the blue emission channel. Images were merged (Fig. 5B), and the numbers of germinated conidia/short hyphae in the whole well were calculated, as explained in Materials and Methods (Fig. 5B). By these means we could calculate that ~2% of the PD-9 spore population and ~1% of the PD-104 conidia were able to germinate at 2× MIC and ~0.35% for both isolates in 3× MIC. Interestingly, addition of GAG enhanced the number of persisters in PD-104 (from 1.09 to 2.80% [$P$ = 0.0267] at 2× MIC and from 0.41 to 0.75% [$P$ = 0.038]) but not in PD-9 (Fig. 5B). Next, we inoculated the four isolates in duplicated lines of 96-well plates containing a high concentration of voriconazole (4 $\mu$g/mL) in all wells and added GAG to one line of each isolate. The entire contents of wells were plated 48 h after inoculation on PDA-rich plates to count the number of viable CFU. As observed before (Fig. 1C), we found that the persister isolates (PD-9 and PD-104) maintained viability of a significant number of conidia for an extended period of time (Fig. 5C). Interestingly, GAG addition seemed to slightly increase the number of grown CFU for all isolates, although this increment was only significant for the PD-104 isolate (PD-104 versus PD-104+GAG [$P$ = 0.0437]).

These results suggest that a high level of GAG can potentiate persistence in at least one persister isolate. Future investigations will examine the underlying mechanism and determine why GAG addition is isolate specific.

**Persistence can be detected in diverse *A. fumigatus* collections of isolates.** To corroborate that certain *A. fumigatus* isolates display persistence to voriconazole, we decided to screen two independent collection of isolates. Initially, we tested an environmental library of isolates collected in the area of Manchester, United Kingdom (86). We screened all 157 isolates for growth under the microscope after 72 h incubation in the presence of a high concentration of voriconazole (8 $\mu$g/mL). We found that 34 isolates were able to show a limited degree of growth, suggesting that they might be persisters (see Table S9 at https://doi.org/10.5281/zenodo.7533742). Next, we employed disc diffusion assays with 0.8 mg/mL voriconazole and observed that 15 of the 34 isolates were able to form colonies in the halo (see Table S9 at https://doi.org/10.5281/zenodo.7533742). Broth dilution assay with the original isolates and the CoHs revealed that 10 of these isolates did not have increased MICs, indicating that they were persister strains (see Table S9 at https://doi.org/10.5281/zenodo.7533742). Therefore, at least ~6% of this collection were persisters to voriconazole. We further evaluated a collection of 17 azole-susceptible clinical isolates obtained from the Tel Aviv University (TAU) Medical Center (see Table S9 at https://doi.org/10.5281/zenodo.7533742) using the disc diffusion assay. We found that 6 of 17 (35%) isolates were able to grow small colonies in the inhibition halo (see Fig. S10A). Colonies picked from within the halo formed upon reinoculation similar size halos, indicating that the MIC had not increased, and grew a similar number of CoHs, suggesting that the isolates are persisters.

Therefore, evaluation of independent collection of isolates seem to always retrieve a significant number of persister strains, demonstrating that this is quite a common phenomenon.

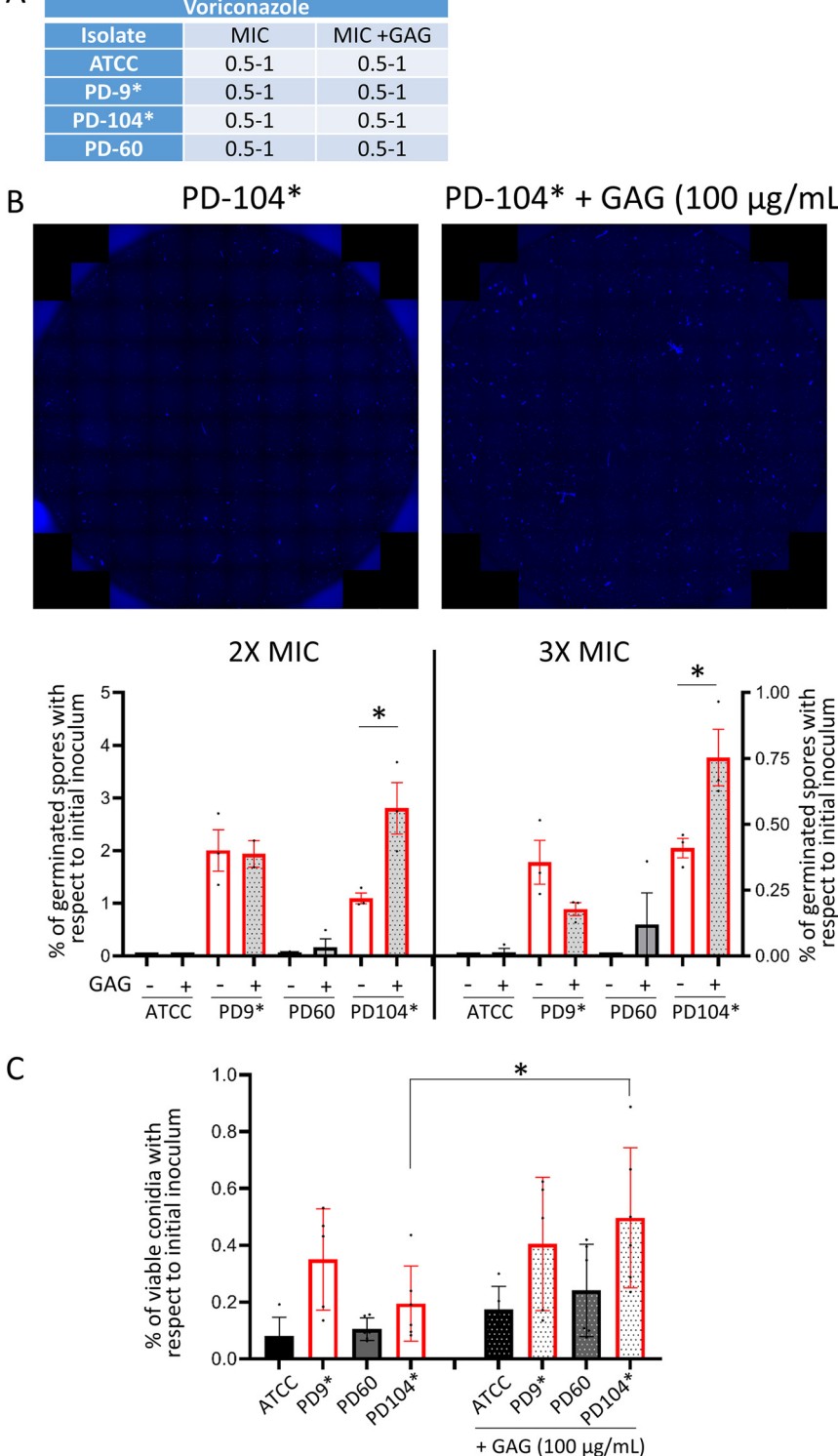

**FIG 5** External addition of GAG potentiates persistence in the isolate PD-104. (A) Addition of GAG does not change the MIC of the isolates. (B) Representative merges of 87 photos to cover the whole well of broth dilution assays. The fungal material was stained with CFW. Representative wells of PD-104 with or without GAG at 2× the MIC are shown. Counting of germinated conidia/short hyphae in entire wells of 2× (left) and 3× (right) MIC of broth dilution assays. External addition of GAG significantly increased the number of germlings in PD-104 from 1.09 to 2.80% ($P = 0.0267$) at 2× MIC and from 0.41 to 0.75% ($P = 0.038$), as assessed using a two-tailed unpaired $t$ test. Two independent experiments with three technical replicates were performed. The graphs represent the means and SD. (C) Plating of the entire content of wells containing 4 $\mu$g/mL of voriconazole shows that external

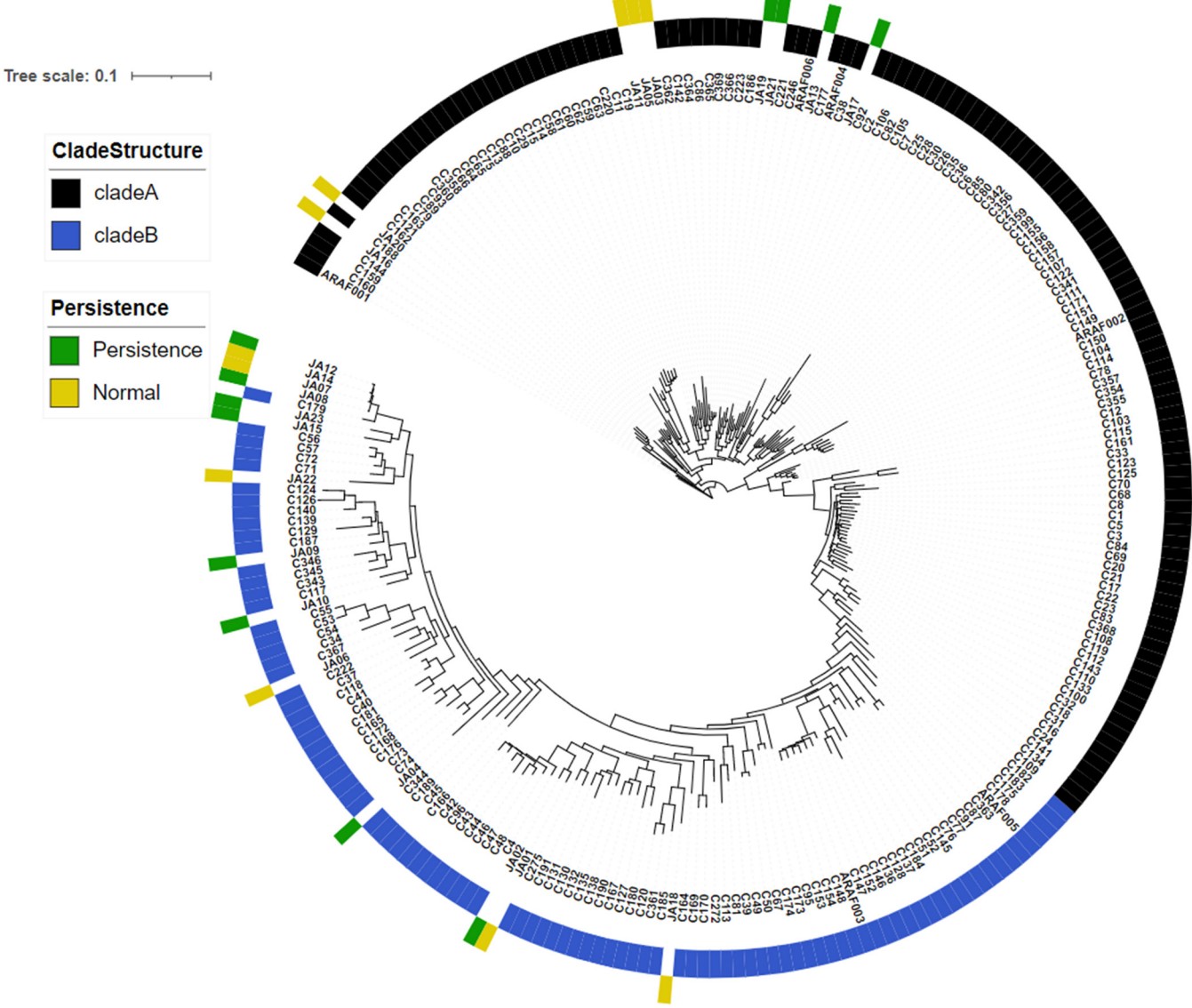

**FIG 6** Persister isolates do not associate in an unique cluster. The genomes of newly sequenced persister and nonpersister isolates are depicted, as characterized in this study (see Table S9 at https://doi.org/10.5281/zenodo.7533742), distributed scattered when included in the previously generated tree (79). Therefore, persistence is not a feature that has evolved in a particular lineage.

**Persistence is not a cluster-specific trait.** Recent studies suggest that the tandem repeats (TRs) in the promoter region of *cyp51A*, which cause high levels of antifungal resistance, possibly have evolved in the environment, due to the use of demethylase inhibitors (azoles) in the fields (87). Interestingly, the isolates with the $TR_{34}/L98H$ polymorphism have been found to be closely related (79, 87), which suggests that this mechanism has evolved in a distinct cluster of the species. To investigate whether the phenomenon of persistence could also be cluster specific, we sequenced the genome of 23 of our isolates, including 12 persisters and 11 nonpersisters (see Table S9 at https://doi.org/10.5281/zenodo.7533742), and integrated their genomes into a recently published phylogenetic analysis (79). We found that both persister and nonpersister isolates are scattered through the entire tree, demonstrating that persistence is not

**FIG 5** Legend (Continued)

addition of GAG significantly increased the number of viable PD-104 cells that can be recovered 48 h after inoculation ($P = 0.0437$, one-way ANOVA with Tukey's multiple comparisons). Two independent experiments with three biological replicates were performed. The graph represents the means and SD. Persister isolates are labeled with an asterisk (*).

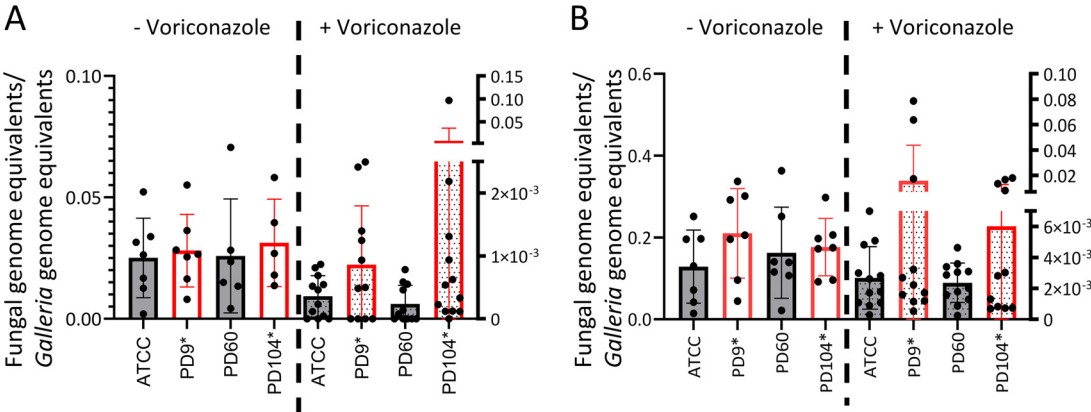

**FIG 7** In some larvae, voriconazole eliminates persister isolates less efficiently than nonpersister strains in a *Galleria mellonella* infection model. The fungal burden was measured by qPCR 72 h after infection with $10^4$ conidia (A) or at 48 h after infection with $5 \times 10^4$ conidia (B) for all strains; the fungal burden was greatly lower in larvae that had received a voriconazole treatment (8 $\mu$g/mL) than in those who had not. The reduction in burden was bigger in nonpersister isolates than in persister isolates, although these differences were not significant. Several treated larvae infected with persister isolates had noticeably higher burdens than others: 4/11 for PD-9 and 3/11 for PD-104 (A) and 3/11 for PD-9 and 4/11 for PD-104 (B). Each graph displays the combined data from two independent experiments. Persister isolates are labeled with an asterisk (*).

the trait of a specific cluster of *A. fumigatus* (Fig. 6). In addition, it is noteworthy that some persister and nonpersister isolates seem to be genetically very similar, since they are close in the tree (JA12-JA14-JA7-JA8 and also JA1-JA2; Fig. 6). This suggests that the genetic features that enable persistence are modest and/or that nongenomic features, as the transcriptional response, are important for isolate specificity.

**In a *Galleria mellonella* infection model, voriconazole treatment seems to be less efficient against persister isolates in some larvae.** To evaluate whether persistence may be relevant during antifungal treatment, we employed the *Galleria mellonella* mini-host model of infection. This model has been successfully used to investigate the efficiency of azole treatment against *A. fumigatus* (88, 89). In addition, a recent study has characterized the pharmacokinetics of voriconazole in infected larvae, which allowed us to select the optimal dose to reach a high concentration of the drug in the hemolymph (above the MICs of the isolates) but that is nearly completely removed in 24 h (90).

Initially, we performed a survival experiment with all four strains to investigate whether they could have different virulence potential. We infected larvae with $10^4$ or $5 \times 10^4$ conidia of nonpersister (ATCC or PD-60) or persister (PD-9 or PD-104) isolates and monitored the mortality for 10 days. All the isolates killed larvae at a similar rate (see Fig. S10B), demonstrating that they are similarly virulent. Therefore, we aimed to use this model to investigate whether persistence could potentially cause treatment failure. We reasoned that a voriconazole treatment should very efficiently eradicate susceptible, nonpersister strains, but it may not be able to eliminate persister strains with the same efficiency in all individuals. We infected larvae with $10^4$ (Fig. 7A) or $5 \times 10^4$ (Fig. 7B) conidia of nonpersister (ATCC or PD-60) or persister (PD-9 or PD-104) isolates, administered one dose of 8 $\mu$g/larva voriconazole or PBS at the time of infection, and measured the fungal burden at 72 h (Fig. 7A) or 48 h (Fig. 7B) after infection. As expected, voriconazole treatment was efficient and dramatically decreased the fungal burden for all isolates (Fig. 7). For voriconazole-treated groups, the mean fungal burden was larger for persister than for nonpersister isolates, although these differences were not significant (ATCC versus PD-9 [$P = 0.41$ for $10^4$ and $P = 0.57$ for $5 \times 10^4$] and ATCC versus PD-104 [$P = 0.27$ for $10^4$ and $P = 0.69$ for $5 \times 10^4$]; [Mann-Whitney test]) due to the high variability in burdens among individuals infected with persister isolates. Indeed, at both infectious doses, several larvae infected with persister isolates had noticeably greater burdens than all the others (Fig. 7), which we speculate might indicate that in some individuals infected with persistent isolates treatment may be less efficient in eliminating the fungus.

## DISCUSSION

The phenomenon of persistence to antimicrobials was first observed in bacteria 80 years ago (91, 92). In the last decade, intensive research has permitted unraveling various underlying mechanisms in diverse bacteria, and it has become clear that persistence can cause treatment failure and lead to the development of resistance (25, 93). However, our insight into antifungal persistence and tolerance in fungal pathogens is still in its infancy.

We have recently shown that all conidia in a tolerant *A. fumigatus* isolate are able to grow at high concentrations of caspofungin (40). In *C. albicans*, tolerance to fluconazole was described to be a subpopulation effect, but the ratio of tolerant cells was reported to be elevated (35). In both cases, tolerance was observed in response to static drugs, so it seems that the phenomenon of tolerance in fungi associates with fungistatic drugs. In *C. albicans*, the formation of amphotericin B (AmB) persister cells has been described (94). Of note, this phenomenon has only been described within biofilms and cannot be found in planktonic cells (95), possibly due to the physical protection conferred by the biofilm, which may be relevant for the capacity of these cells to survive in the presence of AmB. Indeed, in *A. fumigatus* it was shown that the hypoxic microenvironment developed in biofilms led to reduced metabolic activity in the basal biofilm level, leading to the formation of cells that can survive the antifungal challenge and serve as a drug-resistant reservoir (96).

We have observed that a small subpopulation (0.1 to 5%) of certain *A. fumigatus* isolates can survive for extended periods and even grow at low rates in the presence of supra-MICs of the fungicidal drug voriconazole. In bacteria, tolerance and persistence have been classically explained by a downregulation of metabolism and cell cycle, which triggers a status of dormancy that permits survival despite the action of the drug. However, recent research in pathogenic bacteria has demonstrated that active metabolic responses are required for persistence (97, 98), and slow-growing persisters have also been detected *in vitro* (99) and *in vivo* (100). Active metabolism may even be advantageous, as in the case of *Salmonella*, where persisters have been shown to undermine host defenses (101). Interestingly, *C. albicans* AmB persistence has been described to involve not only the downregulation of primary metabolism but also an increase in stress responses and oxidative defensive mechanisms (102). We have observed that active growth is possible at 2- or even 3-fold the MIC and that this growth entails a distinct transcriptional profile. Therefore, this phenomenon is not merely survival of dormant conidia in the presence of the drug but also seems to be an active mechanism that enables a subpopulation of certain isolates to withstand the action of voriconazole for an extended period. Indeed, using resazurin (an oxidation-reduction indicator used for the measurement of metabolic activity and proliferation of living cells, whose use has been optimized for *A. fumigatus* [103]), we could detect a slight metabolic activity at supra-MICs (see Fig. S10C). This might explain why hypoxia is the only environmental condition that reduced persistence, since in low oxygen the metabolic activity is reduced (96) and the energy-generating metabolism changes (104). Intriguingly, our results demonstrate that persister isolates can survive for extended periods at very high concentrations of voriconazole, but active persister growth can only be achieved by certain persister isolates (see Fig. S3) and only takes place at the lower supra-MICs (Fig. 1B and D). Therefore, we hypothesize that different mechanisms are possibly at play for the persistence phenomenon. Besides, we conjecture that this might have implications in the treatment of aspergillosis caused by persister isolates. We found that persister isolates survive approximately twice as long as nonpersisters (Fig. 1E); according to this, we speculate that patients with high, well-maintained, drug levels in blood (ca. 3 to 5 mg/L) should be able to eradicate persister isolates (as long as the treatment is not discontinued), just that it could take longer. However, if the levels are not maintained that high, a persister strain could be able to survive and maybe cause a relapse. Furthermore, in patients reaching classically considered good levels (>1 mg/mL, [105–107]), but not that high (<3 mg/L), an actively growing persister could still be able to grow at low rates and thus cause treatment failure.

The concept of persistence entails two intriguing aspects. First, it is an isolate-dependent phenomenon, which means that there must be a genetic basis that underlies persistence. We have shown that the persister isolates do not belong to a specific lineage,

suggesting that this feature has not appeared as an evolutionary trait in a lineage of isolates. Recently, it has been described that each *A. fumigatus* strain carries a particular set of accessory genes, which associates with different levels of virulence and drug resistance capacities (108). In the same vein, we speculate that the presence or absence of certain accessory genes may enable certain isolates to persist in the presence of azoles. Future research will evaluate whether this hypothesis is true. A second intriguing aspect of persistence is that it is a subpopulation phenomenon, meaning that within an isogenic isolate only a few conidia are able to survive and grow in the presence of the drug. In bacteria, stochastic expression of key genes has often been proposed as an important underlying cause of persistence (81, 82, 109, 110). This includes stochastic high expression of genes that directly confer resistance (30) and efflux activity (111). Similarly, in our RNA-seq analysis we have detected a higher level of expression of genes of the sterol biosynthesis pathway, including *cyp51A*, and of the azole exporter Cdr1B in persister cells. Therefore, it is plausible that stochastic high levels of these genes could generate the capacity to survive for an extended period in the presence of supra-MICs of voriconazole. In addition, we detected a higher level of expression of GAG biosynthetic genes, and we have observed that externally added GAG can increase the number of persisters in the isolate PD-104. Interestingly, this polysaccharide did not increase the isolate's MIC, indicating that it does not provide a physical barrier that shields the fungus or that it cannot somehow degrade the drug. Future investigations will aim to unravel why GAG potentiates persistence in PD-104, and not in PD-9, which is interesting since it suggests again that there likely are multiple persistence mechanisms. GAG is a very relevant *A. fumigatus* virulence factor, with multiple described adhesion and immunosuppressive activities (112, 113), and here we propose that it may have one more important role in potentiating persistence to voriconazole in some isolates.

In bacterial infections, there is significant evidence supporting a role for the phenomena of tolerance and persistence in antibiotic treatment failure (16, 114). In contrast, the knowledge about these phenomena in fungal infections is still very scarce. Probably, the best-studied phenomenon is heteroresistance in *Cryptococcus neoformans* and *C. gattii* (115–117), which has been found to be a major cause of treatment failure in cryptococcal meningitis (118–122). As mentioned above, there is already evidence indicating that fluconazole tolerance can cause treatment failure in invasive candidiasis (17, 35), and limited evidence suggests that caspofungin tolerance may cause treatment failure in *Aspergillus* infections (41). Here, using a *Galleria* model of infection, we have observed that after a voriconazole treatment, several larvae infected with persister isolates had noticeably higher fungal burdens. Therefore, we hypothesize that, in certain individuals, persistence might reduce the efficacy of voriconazole treatment in *A. fumigatus* infections, which, if proven true, could mean that persistence may cause therapeutic failure in some patients. This speculation will be difficult to test, since it implies that the experiments produce intrinsically variable data: in some individuals treatment works (i.e., low burden) and in some it does not (i.e., high burden). These highly variable data make the normal statistical analyses of little value (the high deviation will prevent statistical significance) and therefore will require different approaches and/or types of analysis.

In conclusion, we have presented here a thorough description and characterization of voriconazole persistence in *A. fumigatus*, proposed potential mechanisms, and provided a first evidence to suggest that this persistence might be relevant for the efficacy of treatment. We therefore propose that voriconazole persistence might cause treatment failure in certain individuals. Obviously, much more research is needed to prove or disprove this hypothesis. Thus, we will continue to investigate persistence in *A. fumigatus*, and we also exhort the fungal research community to consider and investigate this phenomenon in order to continue to further clarify this important topic.

## MATERIALS AND METHODS

***A. fumigatus* strains and culture conditions.** All isolates utilized in the course of this study are listed in Tables S1 and S8 in the supplemental material. Briefly, information about the common laboratory strains used can be found in (123), the isolates from Paul Dyer collection are described elsewhere

(124), and the collection of isolates from the area of Manchester are as described previously (86). The third collection of isolates was collected at the TAU Medical Center (Tel-Aviv, Israel), these 17 strains were characterized for their antifungal resistance (all susceptible) profile and *cyp51A* genotype (all wild type) (data not shown). Persister isolates are labeled in all figures with an asterisk (*).

Isolates were routinely grown on potato dextrose agar (PDA; Oxoid) for 72 h to obtain fresh spores for each experiment. All experiments were performed with fresh spores except in the test for persistence with old conidia. *Aspergillus* minimal medium (AMM) was prepared following a standard recipe (125). Sabouraud (Oxoid) and 2% yeast extract glucose (YAG; 0.5% yeast extract, 1.7% agar, $1\times$ trace elements) media were used in specific experiments.

To evaluate persister growth and determine the MIC, isolates were grown on RPMI 1640 (Sigma) with 35 g/L MOPS (morpholinepropanesulfonic acid; Alfa Aesar) and 2% glucose (pH 7).

GAG was obtained from *A. fumigatus* $\Delta ku80$ strain A1160, grown 2 days in a 1.5-L Brian fermenter at room temperature. GAG isolation and purification was carried out as previously described (126). Briefly, the medium supernatant was collected by filtration and was adjusted to pH 3 by the addition of 100 $\mu$L of 12 M HCl per 100 mL of supernatant. Two volumes of precooled ethanol (4°C) were added, and GAG was precipitated for 3 h at 4°C. The precipitate was collected by centrifugation for 20 min at 5,000 $\times$ *g* at 4°C and then washed twice with a 1/10 dilution of the culture volume of 200 mM NaCl for 1 h under agitation (100 rpm). GAG was dialyzed against tap water and twice against purified water (24 h each) and finally lyophilized to dryness and stored at ambient temperature.

To calculate the germination rate, $10^4$ conidia of each strain were inoculated in 200 $\mu$L of liquid RPMI in 96-well plates and imaged with a Nikon TI microscope equipped with a 37°C incubator and a $40\times$ objective, with one picture captured every 30 min using NIS-Elements 4.0 (Nikon) software. The Cell Counter ImageJ platform plug-in (http://rsb.info.nih.gov/ij/index.html) was used to differentially count resting or swollen conidia versus germinated conidia.

To calculate the growth rate on solid medium, $10^3$ conidia of each strain were inoculated on RPMI plates, and the colony diameter was measured in two different angles per colony at 16, 24, 40, and 48 h. To calculate growth rate on liquid medium $2 \times 10^3$ of each strain were inoculated in 200 $\mu$L of liquid RPMI in 96-well plates, and optical density measurements at 600 nm ($OD_{600}$) were obtained every 10 min. Growth curves were analyzed using the R package Bayesian Estimation of Change-points in the Slope of Multivariate Time-Series (BEAST) (127) with the following parameters: burn, 2,000 to 5,000 iterations; zeroNormalization=TRUE; and a blank threshold of 0.095. Confidence intervals of the first change point were plotted using GraphPad Prism.

**Evaluation of persistence and determination of MICs.** To determine persistence using the disc assay, $4 \times 10^4$ conidia of each isolate was evenly spread on a solidified RPMI plate (1.5% agar), and 10 $\mu$L of 0.8 mg/mL voriconazole or 3.2 mg/mL itraconazole (Acros Organics) was added to a Whatman 6-mm antibiotic assay disc, which was placed in the middle of the plate. For *in vitro* treatment with adjuvant and combinatorial drugs, the agents were added to the RPMI medium at the final concentration detailed in each section. Plates were incubated for 5 days at 37°C. Persister colonies were defined as those with a degree of physical separation from the edge of the loan of growth into the halo of inhibition.

MICs were calculated using the broth microdilution method according to EUCAST E.Def. 9.3 instructions (128). A total of $2.5 \times 10^4$ conidia were used in each well.

**Evaluation of the conidial survival to voriconazole fungicidal action.** To calculate the number of conidia that survived after exposure to high concentration of azoles, full contents of the wells containing the highest concentration of each drug (8 $\mu$g/mL) in the broth dilution assays were plated after 48 to 72 h incubation on PDA and further incubated for 24 to 48 h at 37°C.

Microscopic analysis was used to investigate the survival of strains upon withdrawal of voriconazole at the single-cell level. A total of $1.5 \times 10^4$ conidia from each strain were inoculated in 300 $\mu$L of RPMI medium in the presence of 8 $\mu$g/mL voriconazole (European Pharmacopoeia EP Reference Standard; Merck), followed by incubation for 48 h at 37°C with occasional shaking. The strains were then centrifuged at 4,000 rpm for 5 min, the media were discarded, and the conidia were resuspended in 2 mL of filtered PBS. This wash step was repeated three times before the conidia were resuspended in 600 $\mu$L of drug-free RPMI. Then, 300 $\mu$L of each suspension (approximately $7.5 \times 10^3$ conidia) was added into the wells of a 96-well plate. The wells were then imaged for 72 h, every 2 h, on a Nikon Eclipse Ti microscope, using a Nikon CFI Plan Fluor ELWD $20\times$/0.45 NA objective, captured with a Hamamatsu ORCA-FLASH4.0 LT+ camera (Hamamatsu Photonics), and manipulated using NIS-Elements AR 5.11.01 (Nikon). A video was prepared using Fiji (129).

Microscopy to observe growth at supra-MIC conditions was done in a THUNDER Imager Live Cell microscope, with an HC PL FLUOTAR L $20\times$/0.40 DRY objective. Images were captured using a Leica-DFC9000GTC-VSC13067 camera and Las X Leica application suite (v3.7) software.

To construct the killing curve, the isolates were inoculated in 10 mL of RPMI containing 4 $\mu$g/mL voriconazole. Aliquots for each culture were taken at the time of inoculation (100 $\mu$L) and every 24 h (1 mL) thereafter. The aliquots were spun at 16,000 $\times$ *g* for 5 min, resuspended in 1 mL of PBS, and vortexed. This process was repeated twice to wash off the drug. Finally, conidia were resuspended in 1 mL of PBS, and a fraction was plated on PDA plates (50 $\mu$L at 0 h, 100 $\mu$L at 24 h, and 1 mL for 48, 72, and 96 h). The experiment was repeated three times.

To calculate the percentages of germinating conidia, the wells of broth dilution assays containing $2\times$ and $3\times$ MIC and the maximum drug concentration (8 $\mu$g/mL) were stained (after reading the MIC) with 10 $\mu$g/mL of calcofluor white (CFW; Sigma) for 5 min. The entire wells were imaged with a THUNDER Imager Live Cell microscope, using a HC PL FLUOTAR L $20\times$/0.40 DRY objective (with filter conditions EX:375-435/DCC:455/EM:450-490), a Leica-DFC9000GTC-VSC13067 camera, and Las X v3.7 software. Merged images were analyzed using Fiji (129). Briefly, the merged images were converted to 8-bit, and a threshold was set for the images so that the conidia and germlings could be detected over the background (20 to 255). The option

analyze particle was then executed, setting a minimum size of 0.04 square inches (which was found to exclude the resting conidia). Wells containing the maximum concentration of the drug, where only resting conidia can be found, were used to set the background level of detection for each isolate (consisting of aggregates of conidia, impurities, and carryover conidiophores from the inoculum). The percentage of germinated conidia/short hyphae was calculated as follows: (the number of counted particles per well – the number of counted particles in max drug)/25,000 (inoculum).

**Nucleic acid isolation.** For DNA extraction, $10^6$ spores of each *A. fumigatus* isolate were grown overnight on liquid *Aspergillus* minimal medium, filtered through Miracloth paper (Merck), snap-frozen, and ground in the constant presence of liquid nitrogen. Subsequently, DNA was extracted using a standard CTAB (cetyltrimethylammonium bromide) method.

For RNA extraction, *A. fumigatus* isolates were grown on 0.45-$\mu$m-pore size nylon membranes (GE Healthcare) placed on RPMI plates, followed by incubation for 1 day (without drug) or 5 days (with voriconazole in a disc). Mycelia were scratched off using a spatula and immediately frozen and ground in liquid nitrogen. Subsequently, RNA was extracted and purified with a Plant RNeasy minikit (Qiagen) according to the manufacturer's instructions for filamentous fungi and using on-column DNase treatment.

RNA retrotranscription was performed using a SuperScript IV reverse transcriptase kit (Thermo Fisher), with random hexamers and according to the manufacturer's instructions.

For fungal burden calculations in *Galleria mellonella*, DNA was extracted from decapitated larvae using a DNeasy blood and tissue kit (Qiagen) according to the manufacturer's instructions for animal tissue, with overnight incubation at 56°C.

***Galleria mellonella* infection.** Sixth-stage instar larval *G. mellonella* moths were purchased from R. J. Mous Live Bait (Eigen Haard, The Netherlands). Randomly selected groups of larvae (250 to 350 mg) were injected in the last left proleg with 5 $\mu$L of a conidial suspension ($2 \times 10^6$ conidia/mL; $10^4$ conidia/larva) of the correspondent *A. fumigatus* isolate using a Hamilton syringe. At 2 h after infection, relevant groups were injected in the last right proleg with 5 $\mu$L of a suspension containing 1,600 $\mu$g/mL (8 $\mu$g/larvae) voriconazole. A PBS control group was subjected to the same treatment, but without fungal infection.

**RT-PCR.** To detect the fungal burden, 500-ng portions of DNA extracted from each larva were subjected to quantitative PCR (qPCR) using a SensiMic SybR green kit (Bioline). Forward (5′-ACTTCCGCAATGGAC GTTAC-3′) and reverse (5′-GGATGTTGTTGGGAATCCAC-3′) primers were used to amplify the *A. fumigatus* $\beta$-tubulin gene *tubA* (AFUA_1G10910). The primers designed to amplify the elongation factor 1$\alpha$ (Ef-1$\alpha$) were as follows: forward (5′-AACCTCCTTACAGTGAATCC-3′) and reverse (5′-ATGTTATCTCCGTGCCAG-3′). Standard curves were calculated using different concentrations of fungal and larval gDNA pure template. Negative controls containing no template DNA were subjected to the same procedure to exclude or detect any possible contamination. Three technical replicates were prepared for each sample. qPCRs were performed using a CFX96 real-time system (Bio-Rad) with the following thermal cycling parameters: 95°C for 10 min and 40 cycles of 95°C for 15 s, 58°C for 15 s, and 72°C for 15 s. The fungal burden was calculated by normalizing the number of fungal genome equivalents (i.e., the number of copies of the tubulin gene) to the larval genome equivalents in the sample (i.e., the number of copies of the a Ef-1$\alpha$ gene), as we have reported previously (130).

To measure the relative expression of selected genes, 1 $\mu$L of each synthesized cDNA was subjected to RT-qPCR using a SensiMic SybR green kit (Bioline). The primers used to amplify each gene are shown in Table S10 at https://doi.org/10.5281/zenodo.7533742. The efficiency (*E*) of each RT-PCT (primer pair) was calculated according to the following formula: $E = 10^{[-1/slope]}$ (*tubA* = 2.044, *cyp51A* = 2.074, *cdr1B* = 2.092, *mfsC* = 2.043, *sph3* = 2.167, *uge3* = 2.0008, *hxk* = 2.227, and *adg3* = 2.087). Three technical replicates were prepared for each sample. RT-PCRs were performed using a CFX96 real-time system (Bio-Rad) with the following thermal cycling parameters: 95°C for 10 min and 40 cycles of 95°C for 15 s, 58°C for 15 s, and 72°C for 20 s. The relative gene expression was calculated using the Pffafl method (131).

**RNA sequencing.** Total RNA was submitted to the Genomic Technologies Core Facility (GTCF) at the University of Manchester, Manchester, United Kingdom. The quality and integrity of the RNA samples were assessed using 4200 TapeStation (Agilent Technologies), and then libraries were generated using an Illumina Stranded mRNA Prep Ligation kit according to the manufacturer's protocol. Briefly, total RNA (typically, 0.025 to 1 $\mu$g) was used as input material from which polyadenylated mRNA was purified using poly(T), oligonucleotide-attached, magnetic beads. Next, the mRNA was fragmented under elevated temperature and then reverse transcribed into first-strand cDNA using random hexamer primers and in the presence of actinomycin D (thus improving strand specificity while mitigating spurious DNA-dependent synthesis). After removal of the template RNA, second-strand cDNA was then synthesized to yield blunt-ended, double-stranded cDNA fragments. Strand specificity was maintained by the incorporation of dUTP in place of dTTP to quench the second strand during subsequent amplification. Following a single adenine (A) base addition, adapters with a corresponding, complementary thymine (T) overhang were ligated to the cDNA fragments. Pre-index anchors were then ligated to the ends of the double-stranded cDNA fragments to prepare them for dual indexing. A subsequent PCR amplification step was then used to add the index adapter sequences to create the final cDNA library. The adapter indices enabled the multiplexing of the libraries, which were pooled prior to cluster generation using a cBot instrument. The loaded flow-cell was then paired-end sequenced (76+76 cycles, plus indices) on an Illumina HiSeq4000 instrument. Finally, the output data were demultiplexed, and BCL-to-Fastq conversion performed using Illumina bcl2fastq software (v2.20.0.422).

**RNA-seq analysis.** Unmapped paired-reads of 76 bp from the Illumina HiSeq4000 sequencer were checked using a quality control pipeline consisting of FastQC v0.11.3 (http://www.bioinformatics.babraham .ac.uk/projects/fastqc/) and FastQ Screen v0.13.0 (https://www.bioinformatics.babraham.ac.uk/projects/fastq _screen/).

The reads were trimmed to remove any adapter or poor quality sequence using Trimmomatic v0.39

(132); reads were truncated at a sliding 4 bp window, starting 5′, with a mean quality of <Q20, and removed if the final length was <35 bp. Additional flags included "ILLUMINACLIP:./Truseq3-PE-2_Nextera-PE.fa:2:30:10 SLIDINGWINDOW:4:20 MINLEN:35."

The filtered reads were mapped either to the *A. fumigatus* A1163 reference sequence (GCA_000150145.1/ ASM15014v1) downloaded from the Ensembl Genomes Fungi v44 (133) or PD-104 a *de novo*-assembled genome using STAR v5.3a (134). For A1163 reference the genome index was created using the GTF gene annotation also from Ensembl Genomes Fungi v44. For the PD-104 reference GTF gene, annotation was generated as described below. A suitable flag for the read length (–sjdbOverhang 75) was used. "–quantMode GeneCounts" was used to generate read counts in genes.

Subsequently, PD-104 reads were aligned to the recently generated pangenome (79) using Salmon v1.6.0 (additional parameters for salmon quant: –gcBias [corrects for any GC biases in samples] -l ISR [inward/stranded/reverse read pairs]).

Normalization and differential expression analysis was performed using DESeq2 v1.34.0 (135) on R v4.1.2 (R Core Team [https://www.R-project.org/]). Log fold change shrinkage was applied using the lfcShrink function along with the "apeglm" algorithm (136).

The RNA data are stored in Array Express under accession number accession (https://www.ebi.ac.uk/ arrayexpress/experiments/E-MTAB-11547).

**DNA sequencing (Illumina DNA Prep).** Genomic DNA was submitted to the Genomic Technologies Core Facility (GTCF) at the University of Manchester. Sequencing libraries were generated using on-bead tagmentation chemistry with an Illumina DNA Prep Tagmentation kit according to the manufacturer's protocol. Briefly, bead-linked transposomes were used to mediate the simultaneous fragmentation of gDNA (100 to 500 ng) and the addition of Illumina sequencing primers. Next, reduced-cycle PCR amplification was used to amplify sequencing-ready DNA fragments and to add the indices and adapters. Finally, sequencing-ready fragments were washed and pooled prior to paired-end sequencing (151+151 cycles, plus indices) on an Illumina NextSeq500 instrument. Finally, the output data were demultiplexed (allowing one mismatch), and BCL-to-Fastq conversion was performed using Illumina bcl2fastq software (v2.20.0.422). The RAW data for genome sequences, in fastaq.gz format, are deposited in the NCBI Sequence Read Archive (SRA) under accession number PRJNA908647.

**Method for *de novo* genome assembly of genomes (e.g., PD-104).** Unmapped paired-end reads of 149 bp from an Illumina NextSeq500 sequencer were checked using a quality control pipeline consisting of FastQC v0.11.3 (http://www.bioinformatics.babraham.ac.uk/projects/fastqc/) and FastQ Screen v0.13.0 (https://www.bioinformatics.babraham.ac.uk/projects/fastq_screen/).

The reads were trimmed to remove any adapter or poor-quality sequence using Trimmomatic v0.39 (132). Within the first or last 30 bp of a read, bases were removed if the quality was <30. A sliding 2-bp window, starting 5′, scanned the reads and truncated at a window with a mean quality of <Q20. Reads were removed if the final length was <100 bp. Additional flags included: "ILLUMINACLIP:./Truseq3_Nextera-PE.fa:2:30:10 LEADING:30 TRAILING:30 SLIDINGWINDOW:2:20 MINLEN:100."

Genome assembly was performed using Megahit v1.2.9 (137), using "–no-mercy" as the flag. This resulted in 799 contigs (total, 28,659,847 bp; minimum, 202 bp; maximum, 401,152 bp; average, 35,869 bp; $N_{50}$, 97,740 bp). In comparison, A1163 has 55 contigs (total, 29,205,420 bp).

Quast v5.0.2 (138) was used to determine the quality of the assembly. An example of the PD-104 results can be found in Table S11 at https://doi.org/10.5281/zenodo.7533742.

ProtHint v2.5.0 and GeneMark-EP+ v4.68_lic (139) were used for gene annotation. ProtHint was used to generate the evidence for coding genes in combination with fungal protein download from OrthoDb v10. ProtHint generates two main outputs: "prothint.gff," a GFF file with all reported hints (introns, starts, and stops), and "evidence.gff," a high-confidence subset of "prothint.gff," which is suitable for the GeneMark-EP Plus mode. GeneMark-EP+ was used to generate a GTF file of candidate genes for the PD-104 assembly. Additional parameters were set as "–max_intron 200 –max_intergenic 100000 –min_contig = 1000 –fungus."

In order to determine the function of the novel genes, initially a reciprocal best hit strategy was used comparing protein sequence from PD-104 (using the GeneMark Perl script "get_sequence_from_GTF.pl" with the GTF file and assembled contigs) and A1163 (proteins downloaded from Ensembl Genomes Fungi release 51) (http://ftp.ensemblgenomes.org/pub/fungi/release-51/fasta/aspergillus_fumigatusa1163/pep/Aspergillus _fumigatusa1163.ASM15014v1.pep.all.fa.gz). The best match method was run using proteinortho v6.0.31 (140), which by default uses the program Diamond v2.0.13 (141) to identify protein matches ("-*P* = diamond"). The flag "-sim = 1" was set to obtain best reciprocal matches only.

**Construction of unrooted phylogenetic tree.** Prealignment, the reference Af293 genome (142) was masked to remove low-complexity regions using RepeatMasker (http://www.repeatmasker.org/) v4.1.2-p1 and the Dfam repeat database (143), release 3.5. Quality-checked DNA sequencing reads from the recently generated pangenome study (79) were combined with the reads generated from this study and underwent alignment to the masked reference Af293 genome, using BWA-MEM (144) v0.7.17-r1188. Alignment files were then converted to the sorted BAM format using SAMtools (145) v1.10. Variant calling was then performed with GATK HaplotypeCaller (146) v4.0. All variant calls with a genotype quality of <50 were removed, and low-confidence SNPs were converted to missing data if they met one of the follow criteria: DP < 10 || RMSMappingQuality < 40.0 || QualByDepth < 2.0 || FisherStrand > 60.0 || ABHom < 0.9. Each newly generated VCF file was then compressed, indexed, and then merged into a single file using VCFtools (147) v0.1.16. These data were then converted into a relaxed interleaved Phylip format using vcf2phylip (148) v2.8, and RAxML (149) v8.2.12 under rapid bootstrap analysis using 1,000 replicates with the GTRCAT model, was employed to generate maximum-likelihood phylogenies to test whether persistence is lineage-specific. Phylogenetic trees with overlaying metadata were generated using iTOL (150) v6.5.4.

**Data availability.** Transcriptomic data are available in Array Express (https://www.ebi.ac.uk/arrayexpress/experiments/E-MTAB-11547). All raw data are available upon reasonable request.

The RAW data for genome sequences, in fastaq.gz format, are deposited in the NCBI Sequence Read Archive (SRA) under accession number PRJNA908647.

## SUPPLEMENTAL MATERIAL

Supplemental material is available online only.

**SUPPLEMENTAL FILE 1**, PDF file, 2.1 MB.

## ACKNOWLEDGMENTS

We are deeply grateful to Paul Dyer, who kindly gifted to us the collection of isolates that have been used in this study. We thank Diego Megías and Clara Martín at ISCIII for their help with the microscopy. We are also grateful to Michael Bottery for relevant conversations about bacterial tolerance and persistence. We acknowledge the use of the Genomic Technologies Facility (Faculty of Biology Medicine and Health, University of Manchester) for the DNA and RNA sequencing. Help and support from members of the Manchester Fungal Infection Group and the LRIM is greatly appreciated.

J.A. is funded by an Atracción de Talento Modalidad 1 (020-T1/BMD-200) contract of the Madrid Regional Government. J.S. has been funded by a BSAC Scholarship (bsac-2016-0049). C.V. was funded by FAPESP (2108/00715-3 and 2020/01131-5). G.H.G. has been funded by FAPESP (2016/07870-9 and 2021/04977-5), CNPq (301058/2019-9 and 404735/2018-5) and by the NIH/NIAID (grant R01AI153356). S.G. was cofunded by the NIHR Manchester Research Centre and the Fungal Infection Trust.

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
