## [Reviewer comments · Microbiology Spectrum]

Microbiology Spectrum

***Aspergillus fumigatus* can display persistence to the fungicidal drug voriconazole**

Jennifer Scott, Clara Valero, Alvaro Mato-López, Ian Donaldson, Alejandra Roldán, Harry Chown, Norman van Rhijn, Rebeca Lobo-Vega, Sara Gago, Takanori Furukawa, Alma Morogovsky, Ronen Ben-Ami, Paul Bowyer, Nir Osherov, Thierry Fontaine, Gustavo Goldman, Emilia Mellado, Michael Bromley, and Jorge Amich

Corresponding Author(s): Jorge Amich, Instituto de Salud Carlos III Campus de Majadahonda

Review Timeline:

Submission Date:	November 28, 2022
Editorial Decision:	December 14, 2022
Revision Received:	January 17, 2023
Editorial Decision:	January 26, 2023
Revision Received:	January 27, 2023
Accepted:	February 12, 2023

Editor: Alexandre Alanio

Reviewer(s): The reviewers have opted to remain anonymous.

Transaction Report:

DOI: <https://doi.org/10.1128/spectrum.04770-22>

December 14, 2022

Dr. Jorge Amich
Instituto de Salud Carlos III Campus de Majadahonda
Majadahonda
Spain

Re: Spectrum04770-22 (*Aspergillus fumigatus* can display persistence to the fungicidal drug voriconazole)

Dear Dr. Jorge Amich:

Thank you for submitting your manuscript to Microbiology Spectrum.

Two reviewers evaluated carefully your manuscript and think that a significant work still needs to be done especially in quantifying the observations and specifically on Figure 6 to make the manuscript stronger.

I further suggest the authors to find a relevance in the field of medical mycology for this phenomenon as the authors are targeting the heterogeneity of response to voriconazole exposure with the idea that treatment response should be altered. This can be done in different animal models (*Galleria*, mice, ...).

Link Not Available

Sincerely,

Alexandre Alanio

Journals Department
Reviewer comments:

Reviewer #1 (Comments for the Author):

The authors describe a very thoroughly investigated phenomena of trailing or so-called in vitro persistence of *A. fumigatus* upon voriconazole exposure. This is novel and not investigated so systematically before. Although the authors clearly indicate the limitations of this study and do not make any false claims there is a lack of clinical relevance and missed opportunity to link this by describing the clinical isolates better. This and some other comments and suggestions I will explain below point by point;

-The manuscript is large and contains many photos of plates partly via supplemental materials. As a reviewer the persister colonies were not clearly visible in the pdf version but the original files were of good quality. However, certain colonies were pinpointed as persister colonies while others were not. To me as a naïve observer in this respect it was not always obvious why certain colonies were pinpointed and other's not. Maybe the authors can indicate better how this was selected? Are there more persister colonies but not indicated with such arrow? Otherwise why show all these plates and not just a few representative plates?

-In table S1 the clinical isolates are not described, was there indeed a case of treatment failure in these persister isolates and not in the other non-persister isolates? What was the origin of these isolates? From chronic aspergillosis, acute, colonization? Were there any sequential isolates available from the isolates that showed the persistence phenotype? And can this phenotype be reproduced in any isogenic isolates from these patients?

-Also table S1 indicates Resistance or Susceptible, was this based on MIC testing? Or how else and what was used as breakpoints? I am missing some basic information here that should be included since in a supplemental file there is ample of space to add this relevant information.

-Line 215-216, and line 236-237 what can be considered extended periods of time? If these persister isolates would only be found in chronic disease or also in acute disease? Is a few days then extended period or is weeks an extended period? This also relates to the clinical relevance, if this type of treatment failure would only occur after weeks of treatment is the persister type still able to survive? Or in other words does the extended survival have any clinical relevance?

-Line 334, the persister isolates are marker with an asterix throughout the manuscript which is very helpful. It would help the readers to also mark the other IDs with either (vori)conazole R or S as well.

-Line 421, why did the authors only compare on RNA, these colonies from the different zones would be also very suitable for a comparison on DNA level, why hasn't that been done? Just to rule out genetic mutations it would be good to do this.

-Line 462, maybe it is obvious but was A1163 tested for persister phenotype? I see A1160 in the S1 table, is that a direct derivative of A1163? Or if not, why was A1163 used for comparison but not tested directly for persister phenotype?

-Line 593, also here the clinical isolates from the TAU medical centre are not described, what type of disease and could persisters have contributed to treatment failure in these patients? If not, that would also be relevant to know and it would explain a lot for the clinical relevance of the principle of this paper.

-Figure 6, to define a clade within a group of isolates you need a phylogeny that includes an ancestral lineage and all the descendants of that ancestor that is lacking from the current analysis. What is depicted here is an unrooted tree without ancestor, and therefore no lineages or clades can be defined. The unrooted phylogenetic tree as used in this manuscript (and also reference 65) is a type of phylogenetic tree that only describes the relatedness of a group of organisms such as clustering. The authors have two options, either remove all 'lineage' and 'clade' terminology (and maybe use clustering?) or make a rooted tree by including ancestor isolates (outgroup isolates of *A. fumigatus* or close relatives such as *A. oerlinghausensis*) and detect any clade structure (if it is still there).

-Line 951 (<https://www.ebi.ac.uk/arrayexpress/experiments/E-MTAB-11547>) and line 1029 (<https://www.ebi.ac.uk/arrayexpress/experiments/E-MTAB-11547>), going to this webpage the data is not accessible to me as a reviewer.

Reviewer #2 (Comments for the Author):

The study by Scott et al. examines the ability of *A. fumigatus* isolates to display tolerance or persistence to azole antifungals. This species is an important human fungal pathogen. The authors characterize subpopulations of *A. fumigatus* cells which can survive and grow in the presence of supra-MIC concentrations of voriconazole. This is an important step in characterizing the heterogeneity of responses that fungi can display in response to antifungal drugs. Deciphering this heterogeneity is particularly challenging in filamentous fungi. While this is an important phenomenon to be studied, a major drawback to this study is that most of the data presented is qualitative/descriptive rather than quantitative. Therefore, I suggest additional analyses should be performed before publication. Furthermore, some of the statements regarding the importance of the findings and mechanistic understanding should be toned down.

Major comments:

1. The introductory paragraph on tolerance, persistence and heteroresistance (lines 82-99) needs to clarify to which microbes the authors are referring. The mechanisms underlying these phenomena across kingdoms are highly distinct and sometimes not

defined (especially for fungi). The statements made should also be appropriately referenced - there are only 3 references cited in ~20 lines which is not sufficient for a specialized audience.

2. For the most part, the formation of persistent colonies is described in qualitative rather than quantitative terms. It would seem straightforward to count the persistent colonies formed across several replicates to be able to more accurately compare these across strains and media. This is particularly important for Figures 2-3 and for figures where colonies are difficult to identify from disk diffusion pictures.

3. The authors should be cautious with assigning the label of 'heteroresistance' to the isolates examined as these are typically assigned to bacterial/fungal subpopulations which display much larger differences in resistance levels (~8 to 10-fold higher) than those seen here (~2-fold). In the absence of more in-depth mechanistic analyses, perhaps these annotations are premature.

4. Given that only 2 biological replicates were performed for RNA Seq analyses, this data should be validated using qPCR for a subset of genes.

5. I found the discussion very long and highly speculative, therefore this section could be shortened. In addition, the proposed definitions might not entirely work as Rosenberg et al (2018) have also observed tolerance for fungicidal drugs. Perhaps the proposed table should be published as a commentary/opinion rather than in a research study.

6. As the preliminary *Galleria* data does not yet support the relevance for treatment efficacy, some of the statements regarding the importance of this phenomenon for antifungal treatment should be toned down.

7. In the Importance section, the authors state that "we comprehensively show that some isolates display persistence to this fungicidal antifungal and identify various potential mechanisms of action." As the study does not identify potential mechanisms by which these cells survive in the presence of antifungals (or how they arise), this statement should be toned down. The findings are of a descriptive/associative nature therefore this statement is misleading.

Minor comments:

1. Is there a reason for which the authors switch between RPMI / PDA / AMM media in the first part of the study?

2. Scale bars are missing from images in Figs. S4, S7.

3. Figure legends should include the concentrations of the drugs used. For example, Fig. 3 should include adjuvant concentrations.

4. Is there a reason why a new strain (A1160) is used for RNA Seq profiling in Figure 4? This is the first time the strain is used in the study, this should be explained.

5. I am not sure what additional information the functional association analysis using the STRING database (shown in Fig. S8B-C) provides. This is not explained in the text beyond the fact that the genes examined are in the same network. The functions/process affected by these networks or the gene names should be annotated to the figure or described in the text in a way that is informative to the reader. Therefore, it is not clear how and why this analysis points to a "distinct metabolic response".

6. Line 467 - perhaps the authors mean re-aligned rather than "re-annealed".

7. Line 594 - isolates "which were prior shown not to have any mutation in the *cyp51A* gene promoter or ORF." This statement should be appropriately referenced.

8. All methods sections should include the number of biological replicates performed and the statistical analyses used to determine significance.

Staff Comments:

Preparing Revision Guidelines

Please return the manuscript within 60 days; if you cannot complete the modification within this time period, please contact me. If you do not wish to modify the manuscript and prefer to submit it to another journal, please notify me of your decision

immediately so that the manuscript may be formally withdrawn from consideration by Microbiology Spectrum.

Reviewer #1 (Comments for the Author):

The authors describe a very thoroughly investigated phenomena of trailing or so-called *in vitro* persistence of *A. fumigatus* upon voriconazole exposure. This is novel and not investigated so systematically before. Although the authors clearly indicate the limitations of this study and do not make any false claims there is a lack of clinical relevance and missed opportunity to link this by describing the clinical isolates better. This and some other comments and suggestions I will explain below point by point;

We would like to thank the reviewer for the time taken to review our work and express our gratitude for appreciating that our investigation is thorough.

As there was no insight of azole persistence in *A. fumigatus*, the aim of this study was to detect and characterise this phenomenon *in vitro*, and just provide a first piece of evidence (using the *Galleria* model) to propose that it might be important *in vivo*. Therefore, regrettably, we did not collect well-deposited and curated clinical isolates, and these that we have utilized in this study cannot be used to investigate the potential clinical relevance of persistence.

The PD-47 collection is curated in Paul Dyer's laboratory, but the isolates were randomly sent to him from different clinicians and locations around the Globe. Further details of the isolates can be found in Table S1 of the paper cited as the source of the isolates (<https://doi.org/10.3390/jof6040258>). However, this does not include details about the treatment administered nor the outcome as this data was confidential from source.

Limited information about the TAU collection has been added in Table S9. However, as explained below, this is clearly insufficient to draw any conclusion.

We believe it is still too soon to link persistence to clinical relevance, much more evidence will be required for this. We are currently collecting and analysing clinical isolates from the Spanish Reference Centre, and also consecutive isolates from the same infected patients, to be able to address this important question. However, this is ongoing work and will be presented in future publications.

-The manuscript is large and contains many photos of plates partly via supplemental materials. As a reviewer the persister colonies were not clearly visible in the pdf version but the original files were of good quality. However, certain colonies were pinpointed as persister colonies while others were not. To me as a naïve observer in this respect it was not always obvious why certain colonies were pinpointed and other's not. Maybe the authors can indicate better how this was selected? Are there more persister colonies but not indicated with such arrow? Otherwise why show all these plates and not just a few representative plates?

We apologise for causing this misunderstanding. When several CoHs appeared in certain plates, we did not exhaustively label all colonies. We see now that this can cause confusion. Therefore, we now indicate all CoHs considered persisters (some degree of physical separation from the edge, as mentioned in material and methods) with arrows throughout the figures.

-In table S1 the clinical isolates are not described, was there indeed a case of treatment failure in these persister isolates and not in the other non-persister isolates? What was the origin of these isolates? From chronic aspergillosis, acute, colonization? Were there any sequential isolates available from the isolates that showed the persistence phenotype? And can this phenotype be reproduced in any isogenic isolates from these patients?

Adding to the explanation above, none of the PD-47 isolates (or from the TAU collection) are sequential from the same patient.

We are currently analysing sequential isolates from two haematological patients, and we can confirm to the reviewer that we have detected persistence, and it is reproducible for isogenic isolates. Nevertheless, this is ongoing work that we will present in a future publication.

-Also table S1 indicates Resistance or Susceptible, was this based on MIC testing? Or how else and what was used as breakpoints? I am missing some basic information here that should be included since in a supplemental file there is ample of space to add this relevant information.

We thank the reviewer for pointing out this shortage of information. Resistance or susceptibility was determined using the broth microdilution method, following the EUCAST E.Def. 9.3 instructions, and according to the breakpoints defined in Guinea 2020. This is now clearly stated in the table legend, lines 1313-1315.

-Line 215-216, and line 236-237 what can be considered extended periods of time? If these persisters isolates would only be found in chronic disease or also in acute disease? Is a few days then extended period or is weeks an extended period? This also relates to the clinical relevance, if this type of treatment failure would only occur after weeks of treatment is the persister type still able to survive? Or in other words does the extended survival have any clinical relevance?

We thank the reviewer for this thoughtful comment. It is indeed a key point to understand the potential clinical relevance of persistence. We attempted to discuss this point in lines 605-613.

Looking at the survival curve (Figure 1E), non-persisters are almost killed in 2 days and completely eradicated in 3, whilst persisters maintain viability up to 4 days. This means around double time of survival. We have added this information in line 607. Importantly, the killing dynamics *in vivo* may be different from the dynamics *in vitro*, so this time-scale needs to be considered with care.

To what extent this extended period is relevant in patients that have optimal drug levels will need to be carefully evaluated. However, more importantly, we believe that this might cause problems in certain individuals if the drug does not reach optimal levels and/or if the levels are not well maintained. As mentioned above, we discuss all this in lines 605-613.

Nevertheless, it is important to remark that at this time all this is still speculative. We would continue investigating this phenomenon with the aim to clarify the clinical relevance of persistence, and hope that other groups around the globe decide to test these hypotheses as well.

-Line 334, the persister isolates are marked with an asterisk throughout the manuscript which is very helpful. It would help the readers to also mark the other IDs with either (vori)conazole R or S as well.

We thank the reviewer for this suggestion. We believe that labelling all the susceptible strains would complicate the figures, (additionally, in most of the figures only the strains ATCC and PD-60 appear, and their MICs are shown in Fig. 2D). We have labelled the resistant isolate PD-256 with an "R" superscript. We do not dare to label PD-266 as we propose that it might be

heteroresistant, but Reviewer 2 has pointed out that we should be cautious with assigning this classification.

-Line 421, why did the authors only compare on RNA, these colonies from the different zones would be also very suitable for a comparison on DNA level, why hasn't that been done? Just to rule out genetic mutations it would be good to do this.

We thank the reviewer for this suggestion. We would not expect to normally see any (relevant) difference at the genomic level between the colonies of the halo and the original isolate. We repeatedly picked CoH of PD-9 and PD-104 and re-inoculated them on broth dilution and disc assays, and we did not detect any phenotypic difference that could suggest a genetic modification. The phenotype seems stable and repetitive, therefore suggesting that it is unlikely that it is caused via mutations in the CoHs. In the future, it would be interesting to aiming to evolve persistence in a non-persister isolate and sequence the genome to detect the change(s) that explain it. However, we are yet uncertain if this is possible in *A. fumigatus*, as we also speculate that the presence/absence of accessory genes might be the basis for persistence. All these genetic and genomic analyses are planned for the future.

-Line 462, maybe it is obvious but was A1163 tested for persister phenotype? I see A1160 in the S1 table, is that a direct derivative of A1163? Or if not, why was A1163 used for comparison but not tested directly for persister phenotype?

A1160 is indeed a direct derivative of A1163, and both are derivatives of CEA10. Actually, we tested persistence for the three isolates, and observed an identical phenotype. We just show one of them in Fig. S1, and we selected CEA10 as it is the original isolate. We did not include A1163 in the table to prevent redundancy.

-Line 593, also here the clinical isolates from the TAU medical centre are not described, what type of disease and could persisters have contributed to treatment failure in these patients? If not, that would also be relevant to know and it would explain a lot for the clinical relevance of the principle of this paper.

We have gathered some information of the isolates contained in the "TAU collection". These strains are actually divided in two sets of isolates. The ones that start with "A" are very old, and we only have partial information about them. The only numbered isolates are more recent and we have included the details in Table S9. As it can be seen there, only one isolate belongs to a lethal infection (Invasive Aspergillosis), and in this case, it caused treatment failure. This would agree with a clinical relevance of persistence, but of course we cannot draw any conclusion based on one isolate.

-Figure 6, to define a clade within a group of isolates you need a phylogeny that includes an ancestral lineage and all the descendants of that ancestor that is lacking from the current analysis. What is depicted here is an unrooted tree without ancestor, and therefore no lineages or clades can be defined. The unrooted phylogenetic tree as used in this manuscript (and also reference 65) is a type of phylogenetic tree that only describes the relatedness of a group of organisms such as clustering. The authors have two options, either remove all 'lineage' and 'clade' terminology (and maybe use clustering?) or make a rooted tree by including ancestor

isolates (outgroup isolates of *A. fumigatus* or close relatives such as *A. oerlinghausensis*) and detect any clade structure (if it is still there).

We are deeply thankful to the reviewer for this correction. We have substituted all terms by “cluster” as suggested. We merely intended to study if persistence was a feature of related isolates or not, thus we believe this type of tree is sufficient to support our conclusion.

-Line 951 (<https://www.ebi.ac.uk/arrayexpress/experiments/E-MTAB-11547>) and line 1029 (<https://www.ebi.ac.uk/arrayexpress/experiments/E-MTAB-11547>), going to this webpage the data is not accessible to me as a reviewer.

Indeed, this data will be made publicly available once the manuscript is accepted. Reviewers using the following login can access it:

Username: Reviewer_E-MTAB-11547 / Password: fjxwwiwm

We apologise for not providing this login information before.

This is also true for the genome sequences, which have been deposited in NCBI’s Sequence Read Archive (SRA), as mentioned now in lines 847-848. To access this database as a reviewer, please, see the following link:

<https://dataview.ncbi.nlm.nih.gov/object/PRJNA908647?reviewer=4g8m1ji6dv7ae84p452m74n0nc>

Reviewer #2 (Comments for the Author):

The study by Scott et al. examines the ability of *A. fumigatus* isolates to display tolerance or persistence to azole antifungals. This species is an important human fungal pathogen. The authors characterize subpopulations of *A. fumigatus* cells which can survive and grow in the presence of supra-MIC concentrations of voriconazole. This is an important step in characterizing the heterogeneity of responses that fungi can display in response to antifungal drugs. Deciphering this heterogeneity is particularly challenging in filamentous fungi. While this is an important phenomenon to be studied, a major drawback to this study is that most of the data presented is qualitative/descriptive rather than quantitative. Therefore, I suggest additional analyses should be performed before publication. Furthermore, some of the statements regarding the importance of the findings and mechanistic understanding should be toned down.

We would like to thank the Reviewer for the positive evaluation of our work. We have aimed to address all the comments raised. In particular:

-We have used all our replicates to present quantifications of the colonies of the halo that appeared in the different conditions.

-We have been careful not to overstate our statements or conclusions.

Major comments:

1. The introductory paragraph on tolerance, persistence and heteroresistance (lines 82-99) needs to clarify to which microbes the authors are referring. The mechanisms underlying these phenomena across kingdoms are highly distinct and sometimes not defined (especially for fungi). The statements made should also be appropriately referenced - there are only 3 references cited in ~20 lines which is not sufficient for a specialized audience.

We are grateful for this comment. We initially intended to provide an overarching view of these phenomena in both bacteria and fungi, but we agree with the reviewer that they differ across kingdoms and are not well defined in fungi, thus the definitions were not appropriate for all. We clarify now that we are referring to bacteria (we have adjusted some statements to make them accurate for bacteria), and have added numerous references to support the statements.

2. For the most part, the formation of persistent colonies is described in qualitative rather than quantitative terms. It would seem straightforward to count the persistent colonies formed across several replicates to be able to more accurately compare these across strains and media. This is particularly important for Figures 2-3 and for figures where colonies are difficult to identify from disk diffusion pictures.

We are thankful for this observant comment. We have counted the colonies of the halo present in all the replicates performed in the course of the study, which has allowed us to report quantifications of the CoHs in the different strains and conditions. This undoubtedly improves the study and allows a better understanding of the factors that modulate persistence.

3. The authors should be cautious with assigning the label of 'heteroresistance' to the isolates examined as these are typically assigned to bacterial/fungal subpopulations which display much

larger differences in resistance levels (~8 to 10-fold higher) than those seen here (~2-fold). In the absence of more in-depth mechanistic analyses, perhaps these annotations are premature.

We thank the reviewer for this consideration. To be more cautious, we have denoted the isolate PD-266 as “potentially heteroresistant” throughout the manuscript.

4. Given that only 2 biological replicates were performed for RNA Seq analyses, this data should be validated using qPCR for a subset of genes.

The reviewer is correct. We usually consider that RT-PCR is not required to validate RNA-seq data, but given that we only have 2 biological replicates, this is a good additional control.

We have checked the expression of six selected genes detected as differentially expressed in the RNA-seq by RT-PCR (line 425). As can be seen in Figure S8B, the relative expression of the genes (calculated using the $2^{-\Delta\Delta CT}$ method, as we describe now in lines 783-790) agrees with the data of the RNA-seq.

Additionally, we have taken the chance to validate the higher expression of *agd3* (GAG biosynthesis) in persister growth (lines 481-482).

5. I found the discussion very long and highly speculative, therefore this section could be shortened. In addition, the proposed definitions might not entirely work as Rosenberg et al (2018) have also observed tolerance for fungicidal drugs. Perhaps the proposed table should be published as a commentary/opinion rather than in a research study.

We believe that a highly speculative discussion is appropriate for this study. We present and characterise the phenomenon of persistence, but many questions about its mechanistic basis and its clinical relevance remain open. We consider important to raise these questions and present our thoughts on the topic.

Nevertheless, we accept the reviewer’s criticism that the proposed definitions for persistence might fit better in a commentary/opinion manuscript. Consequently, we have deleted the correspondent paragraph and Table 3. In this manner, the discussion section is shortened.

6. As the preliminary *Galleria* data does not yet support the relevance for treatment efficacy, some of the statements regarding the importance of this phenomenon for antifungal treatment should be toned down.

We completely agree with the reviewer that the *Galleria* data is preliminary. We have been very careful not to overstate the significance of this result.

-Abstract: “seemed to reduce” (line 46)

-Importance: “we provide initial evidence to suggest” (line 60)

-Results: “we speculate might indicate that in some individuals infected with persistent isolates treatment may be less efficient in eliminating the fungus” (lines 562-563)

-Discussion: “we hypothesise that in certain individuals persistence might reduce...” (line 648) and “provided the very first piece of evidence to suggest that it might be relevant” (lines 656-657).

We believe that these statements do not exaggerate the relevance of our result. Please, let us know if we should tone down any of them specifically.

7. In the Importance section, the authors state that "we comprehensively show that some isolates display persistence to this fungicidal antifungal and identify various potential mechanisms of action." As the study does not identify potential mechanisms by which these cells survive in the presence of antifungals (or how they arise), this statement should be toned down. The findings are of a descriptive/associative nature therefore this statement is misleading.

We have substituted "identify" for "propose", to clarify that the potential mechanisms are not validated. (lines 59 and 656).

Minor comments:

1. Is there a reason for which the authors switch between RPMI / PDA / AMM media in the first part of the study?

We routinely use RPMI, as this medium is standardized for the study of *A. fumigatus* resistance (according to EUCAST). With the purpose of introducing the detection of persistence in the diagnosis routine in the future, we believe that this should be the reference medium.

In particular experiments, we made use of a minimal medium (AMM) and rich media because the nutritional environment was described to affect *C. albicans* (Rosenberg et al, 2018 DOI: 10.1038/s41467-018-04926-x) and *S. cerevisiae* (Yu et al, 2022, DOI: 10.1038/s41564-022-01072-5) tolerance. We wanted to check if this was also true for *A. fumigatus* persistence, as we indeed observed. We have modified the sub-heading title and added a line at the beginning of the paragraph to make this clear (lines 251 and 272-274).

2. Scale bars are missing from images in Figs. S4, S7.

We thank the reviewer for noticing this oversight. We have added the scale bars.

3. Figure legends should include the concentrations of the drugs used. For example, Fig. 3 should include adjuvant concentrations.

We agree with the reviewer that the concentration of each drug or compound should be visible in the figure. Accordingly, we have included the concentrations for Fig. 3 in the figure itself, which we believe is more visual.

4. Is there are reason why a new strain (A1160) is used for RNA Seq profiling in Figure 4? This is the first time the strain is used in the study, this should be explained.

We thank the reviewer for this comment. The A1160 strain was initially screened for persistence and, although it is not shown in Fig. S1 (because it is a derivative of CEA10, and consequently showed exactly the same phenotype) it is mentioned in Table S1. We decided to use the A1160 isolate because its genome is well-annotated, whilst the genome of ATCC46645 is not. We reasoned that comparing the transcriptome of our persister isolate with the one of a non-

persister with well-annotated genome would facilitate the analysis. We have included a sentence in line 376 to explain this.

5. I am not sure what additional information the functional association analysis using the STRING database (shown in Fig. S8B-C) provides. This is not explained in the text beyond the fact that the genes examined are in the same network. The functions/process affected by these networks or the gene names should be annotated to the figure or described in the text in a way that is informative to the reader. Therefore, it is not clear how and why this analysis points to a "distinct metabolic response".

To address this fair criticism, we have further analysed the data. We have performed a GO enrichment analysis with the 17 proteins of the interconnected node and detected three enrichments. As proteins in the same node are expected to be biologically related, this suggests that these related proteins participate in those processes, which constitute an active response during persistence growth (lines 448-452). We have added the function of the gene in the figure S8B to facilitate its understanding (the function of genes in Fig, S8C. is detailed in Table 2)

With respect to S8C, we acknowledge that the STRING run does not add new information to the previous analysis; however, we believe it can serve to corroborate the importance of the ergosterol biosynthetic pathway in the persister response.

6. Line 467 - perhaps the authors mean re-aligned rather than "re-annealed". Yes, thank you for pointing out this typo.

7. Line 594 - isolates "which were prior shown not to have any mutation in the *cyp51A* gene promoter or ORF." This statement should be appropriately referenced. We are sorry for this misunderstanding, this result is not published. We mean that before studying the persistence phenotype with these isolates, it was corroborated that none of them carried a point- mutation in the *cyp51A* gene which could affect the voriconazole response. To prevent confusion we have deleted this sentence from the results, and clarified it in material and methods (lines 669-670).

8. All methods sections should include the number of biological replicates performed and the statistical analyses used to determine significance.

We include that information in the figure legends. We believe that this location is better, as the reader can fully understand the experiment and its analysis at the same time as is looking at the results.

Nevertheless, if the Microbiology Spectrum editors require that this information should be in the material and methods we will change it.

January 26, 2023

Dr. Jorge Amich
Instituto de Salud Carlos III Campus de Majadahonda
Majadahonda
Spain

Re: Spectrum04770-22R1 (*Aspergillus fumigatus* can display persistence to the fungicidal drug voriconazole)

Dear Dr. Jorge Amich:

I would like to thank the authors for their straightforward and adapted replies.

A minor point could be fixed on the RTqPCR data to clearly show robust data.

The authors should calculate the expression using RTqPCR with the method integrating the efficiency of each qPCR tested as compared to the reference ones using the pfafl method (Pfafl NAR 2001). Ideally several reference gene should be included in the analysis by using their geometric mean.

1. Vandesompele J, Preter K de, Pattyn F, Poppe B, Roy N van, Paepe A de, Speleman F. 2002. Accurate normalization of real-time quantitative RT-PCR data by geometric averaging of multiple internal control genes. *Genome biology* 3:RESEARCH0034.

1. Pfaffl MW. 2001. A new mathematical model for relative quantification in real-time RT-PCR. *Nucleic Acids Res* 29:e45-e45.

Link Not Available

Sincerely,

Alexandre Alanio

Journals Department
Reviewer comments:

Staff Comments:

Preparing Revision Guidelines

Please return the manuscript within 60 days; if you cannot complete the modification within this time period, please contact me. If you do not wish to modify the manuscript and prefer to submit it to another journal, please notify me of your decision immediately so that the manuscript may be formally withdrawn from consideration by Microbiology Spectrum.

I would like to thank the authors for their straightforward and adapted replies. A minor point could be fixed on the RTqPCR data to clearly show robust data. The authors should calculate the expression using RTqPCR with the method integrating the efficiency of each qPCR tested as compared to the reference ones using the pffafI method (PffafI NAR 2001). Ideally several reference genes should be included in the analysis by using their geometric mean.

1. Vandesompele J, Preter K de, Pattyn F, Poppe B, Roy N van, Paepe A de, Speleman F. 2002. Accurate normalization of real-time quantitative RT-PCR data by geometric averaging of multiple internal control genes. *Genome biology* 3: RESEARCH0034.

1. Pfaffl MW. 2001. A new mathematical model for relative quantification in real-time RT-PCR. *Nucleic Acids Res* 29:e45-e45.

We would like to take the opportunity to thank the Editor for the impeccable managing of our manuscript, as well as for his useful recommendations for the science. We also appreciate his kind words about our replies in the previous revision.

The Editor is correct that the PffafI method is more exact than the $2^{-\Delta\Delta Ct}$. We considered sufficient to use this last method because the efficiencies of amplification were >95% for all primer pairs. Nevertheless, as the efficiency values for all primer pairs are calculated (shown in lines 785-787), we have followed the Editor's recommendation and re-calculated all fold changes using the PffafI method (which is mentioned now in line 790). These more accurate calculations reflect the same patterns of increased/reduced expression as described before, which are in full agreement with the RNA-seq data (Fig. S8B). Therefore, we consider that the RNA-seq data is well validated, which is the ultimate aim of this RT-PCR analysis.

For this reason, we consider that it is not necessary to add more reference genes to this analysis. The patterns of expression are in agreement with the RNA-seq, so even if the exact level of FC could be more precise if more reference genes were added, we believe that it is not required as the aim of the analysis is already fulfilled.

February 12, 2023

Dr. Jorge Amich
Instituto de Salud Carlos III Campus de Majadahonda
Majadahonda
Spain

Re: Spectrum04770-22R2 (*Aspergillus fumigatus* can display persistence to the fungicidal drug voriconazole)

Dear Dr. Jorge Amich:

Your manuscript has been accepted, and I am forwarding it to the ASM Journals Department for publication. You will be notified when your proofs are ready to be viewed.

Sincerely,

Alexandre Alanio
Editor, Microbiology Spectrum
